# Inhibitory neurotransmission drives endocannabinoid degradation to promote memory consolidation

Christophe J. Dubois [1,3,4], Jessica Fawcett-Patel[1,4], Paul A. Katzman[1] & Siqiong June Liu [1,2✉]

Endocannabinoids retrogradely regulate synaptic transmission and their abundance is controlled by the fine balance between endocannabinoid synthesis and degradation. While the common assumption is that "on-demand" release determines endocannabinoid signaling, their rapid degradation is expected to control the temporal profile of endocannabinoid action and may impact neuronal signaling. Here we show that memory formation through fear conditioning selectively accelerates the degradation of endocannabinoids in the cerebellum. Learning induced a lasting increase in GABA release and this was responsible for driving the change in endocannabinoid degradation. Conversely, Gq-DREADD activation of cerebellar Purkinje cells enhanced endocannabinoid signaling and impaired memory consolidation. Our findings identify a previously unappreciated reciprocal interaction between GABA and the endocannabinoid system in which GABA signaling accelerates endocannabinoid degradation, and triggers a form of learning-induced metaplasticity.

[1] Department of Cell Biology and Anatomy, LSU Health Sciences Center, New Orleans, LA 70112, USA. [2] Southeast Louisiana VA Healthcare System, New Orleans, LA 70119, USA. [3] Present address: Univ. Bordeaux, CNRS, EPHE, INCIA, UMR 5287, F-33000 Bordeaux, France. [4] These authors contributed equally: Christophe J. Dubois, Jessica Fawcett-Patel. ✉email: sliu@lsuhsc.edu

Neuromodulators control both synaptic transmission and the intrinsic excitability of neurons and their dysfunction can lead to neurological disorders. Their abundance is controlled by a fine balance between production and degradation. While the common assumption is that neural activity promotes neuromodulator production to influence signaling, rapid chemical degradation process likely controls the temporal profile of neuromodulator action to optimize information processing within a neuronal circuit. Thus in theory, a change in degradation rate would impact neuromodulator signaling, but whether this form of plasticity exists has not been determined. This is surprising given the therapeutic potential of regulating the rate of degradation, as illustrated by the use of cholinesterase inhibitors which prolong the actions of acetylcholine and are used for the treatment of cognitive deficits[1].

Endocannabinoids, such as 2-arachidonoylglycerol (2-AG), are produced when neurons are activated ("on-demand"), and suppress neurotransmitter release and intrinsic excitability[2,3]. Monoacylglycerol lipase (MAGL), a 2-AG degrading enzyme, terminates their activity[4,5]. This process is important as inhibition of endocannabinoid degradation can reduce anxiety-like behavior in rodents[6–8], and stress and alcohol abuse can alter the level of 2-AG degrading enzymes[9,10]. However, whether there is a physiological regulation of degradation rate is not known. We have therefore investigated whether the activity of inhibitory interneurons controls 2-AG degradation in the cerebellar circuitry.

Growing evidence indicates that the cerebellum has non-motor functions and is critical for associative fear conditioning, in particular the consolidation of fear memory[11–13]. Fear conditioning enhances excitatory postsynaptic responses at the parallel fiber—Purkinje cell synapse and presynaptic GABA release from molecular layer interneurons (MLIs, basket and stellate cells), as well as feed-forward inhibitory connectivity in the cerebellum[14–16]. These changes may drive memory consolidation; however, the mechanisms underlying neural plasticity are unknown. Within the cerebellar circuitry, inhibitory interneurons control Purkinje cell activity, and importantly both gamma-Aminobutyric acid release and interneuron activity are suppressed by endocannabinoids. While compelling evidence shows that endocannabinoid production and receptor activation are required for memory extinction[17,18], their role in memory consolidation remains controversial. The observation that 2-AG degrading enzyme inhibitors produce memory deficits raises the possibility that suppression of endocannabinoid signaling might be important for fear memory formation[17–23].

Here we show that associative fear conditioning enhances endocannabinoid degradation. This is *induced* by an increase in the activity of inhibitory interneurons in the cerebellum and is *maintained* by a learning-induced increase in GABA release. Learning selectively elevates 2-AG degrading enzyme levels in vermal lobules V/VI which are involved in associative fear conditioning, leading to a reduction in tonic 2-AG levels. At a behavioral level, we have found that activation of Gq DREADD (Designer Receptor Exclusively Activated by Designer Drugs) in cerebellar Purkinje cells after learning elevated endocannabinoid tone and impaired memory retention. Administration of a CB1R antagonist prevented the memory deficit. Our results reveal a previously unappreciated reciprocal interaction between GABA and the endocannabinoid system in which GABA release accelerates endocannabinoid degradation, which plays an important role in memory consolidation.

## Results

**Fear conditioning accelerates 2-AG degradation**. In the cerebellum, 2-AG is the major endocannabinoid released from Purkinje cells and MLIs (stellate/basket cells). Activation of endocannabinoid receptor 1 (CB1Rs) on the presynaptic terminals of GABAergic stellate cells and glutamatergic granule cells suppresses neurotransmitter release[4,24,25]. 2-AG is removed via degradation by MAGL, which accelerates the recovery of synaptic transmission[4,26]. Thus, a depolarization-induced suppression of neurotransmitter release was used as a functional assay to assess endocannabinoid signaling, as deletion or inhibition of DAGL (diacylglycerol lipase, a rate-limiting enzyme in the production of 2-AG) and CB1R reduces the amplitude of the suppression[27], whereas deletion or inhibition of degradation enzymes prolongs the recovery rate[4,28,29]. These experiments were conducted in cerebellar slices in lobules V/VI where acoustic and nociceptive stimuli converge[30].

Depolarization of stellate cells induced a transient suppression of evoked EPSC (excitatory postsynaptic current) amplitude (DSE, depolarization-induced suppression of excitation) at parallel fiber to stellate cell synapses, which was abolished by a neutral CB1R antagonist, NESS⁰³²⁷ (Fig. 1a, b, g). Thus, the suppression of glutamate release requires CB1R activation and is mediated by endocannabinoids. The effects of fear conditioning were compared with naïve and unpaired controls (Fig. 1c). In naïve animals, depression of EPSCs recovered with a time constant of $8.8 \pm 1.6$ s, and application of a MAGL inhibitor, JZL[184], prolonged the recovery time to $14.3 \pm 1.5$ s (Fig. 1e, f; $P < 0.05$). This result suggests that degradation of 2-AG reduced the duration of DSE, consistent with previous reports[4,5]. Fear conditioning did not alter the magnitude of the peak suppression when tested 15 h after acquisition (Unpaired controls $-58 \pm 5\%$, Fear conditioned $-45 \pm 2\%$, $P > 0.05$, Fig. 1d, g). However, the recovery time of DSE was markedly faster after fear conditioning ($4.2 \pm 0.6$ s), compared to unpaired control animals ($8.7 \pm 0.8$ s, $P < 0.01$; Fig. 1d, f). Both the size and time course of DSE in animals subject to unpaired conditioning were similar to those in naïve animals ($P > 0.05$, Fig. 1b, d, f) indicating that associative fear conditioning, rather than stress, accelerated DSE recovery rate. While evidence supports the idea that degradation enzymes accelerate the time course of DSE in cerebellar neurons, other factors can also influence the recovery time as DSE still recovers slowly when MAGL is deleted or inhibited[4,5]. We reasoned that if learning accelerates DSE recovery via a MAGL-independent mechanism, the recovery time should remain faster in conditioned compared to control mice in the presence of a MAGL inhibitor. In contrast, we found that inhibition of MAGL abolished the difference in the recovery rate of DSE, resulting in a greater increase in recovery time after fear conditioning (paired: 295% vs naïve: 61%, $P < 0.05$, Fig. 1e, f). This indicates that the change in recovery rate of DSE after fear learning is mostly due to an increase in MAGL activity, rather than a MAGL-independent mechanism.

MAGL is present in granule and Bergmann glial cells, and deletion of MAGL prolongs the recovery time of both DSE at excitatory synapses, and DSI (depolarization-induced suppression of inhibition) at cerebellar inhibitory synapses[4]. Thus, an increase in MAGL activity is expected to accelerate the recovery time of DSE as well as of DSI. To test whether learning-induced changes are synapse-specific, we depolarized a postsynaptic stellate cell and monitored inhibitory transmission evoked by stimulating another stellate cell (Fig. 2a). While a suppression of the amplitude of evoked inhibitory postsynaptic currents (IPSCs) (DSI) in conditioned mice was comparable to that in naïve mice, the recovery time of DSI in conditioned mice ($16.0 \pm 1.4$ s) was substantially accelerated relative to naïve controls ($31.3 \pm 2.9$ s; $P < 0.001$, Fig. 2b, c). Therefore, learning markedly increased the recovery rate of depolarization-induced suppression at both excitatory and inhibitory synapses without affecting the

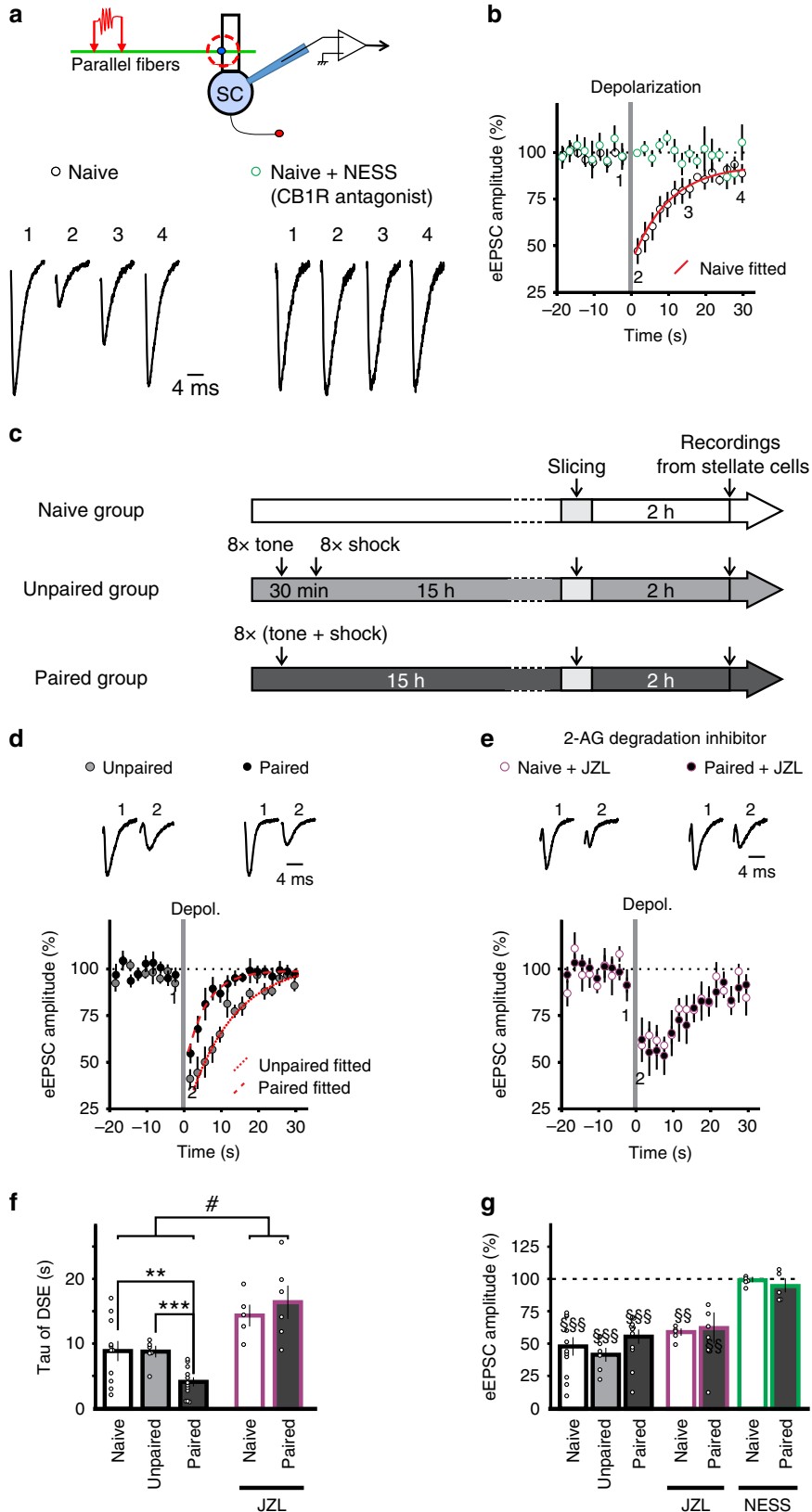

amplitude of the depression. This is consistent with a change in 2-AG degradation that has been shown to alter recovery time and to be synapse-independent[4]. A longer recovery time of DSI than DSE could arise from a rapid removal of 2-AG by MAGL in the presynaptic parallel fibers and processes of Bergmann glial cells at

excitatory synapses, whereas a lack of presynaptic MAGL in MLIs prolongs the clearance of 2-AG at inhibitory synapses.

An increase in eCB degradation is predicted to accelerate the DSI recovery time independent of the type of cells that release 2-AG, whereas an alteration in 2-AG production is likely to be cell

**Fig. 1 Fear conditioning shortens depolarization-induced suppression of excitation (DSE) at the parallel fiber-stellate cell synapse. a** Top, schematic of the experimental procedure. Bottom, sample traces before and after a 2-s depolarization of stellate cells to 0 mV that suppressed the amplitude of EPSCs evoked by PF stimulation (left). This suppression of eEPSC amplitude was blocked by the CB1 receptor antagonist NESS[0327], indicating that endocannabinoids mediate the suppression. **b** Time course of eEPSCs in stellate cells from naïve mice in the presence ($n = 5$ cells) or in the absence ($n = 11$ cells) of NESS[0327]. Best-fit exponential curve (red line) for the recovery phase in the absence of NESS[0327]. **c** Experimental protocols for fear conditioning experiments. **d** EPSC traces (top) and time course in stellate cells from conditioned (paired $n = 12$ cells) and unpaired ($n = 6$ cells) control mice. **e** JZL[184], a MAGL (monoacylglycerol lipase) inhibitor was applied (naïve, $n = 5$ cells; paired $n = 6$ cells). **f, g** Recovery rate constant and peak suppression of eEPSCs during DSE (NESS[0327], paired $n = 5$ cells). **f** Fear conditioning shortened DSE, and inhibition of MAGL prolonged the time course and abolished the difference between the recovery time in paired and naïve control (two-way ANOVA, #, $P = 0.034$; followed by Tukey's post hoc test, **, $P = 0.011$). The unpaired group in the absence of JZL[184] was compared with the paired group using a two-sided unpaired $t$-test (***, $P < 0.001$). **g** Summary of the magnitude of DSE (two-sided paired $t$-tests, §§, $P < 0.01$, §§§, $P < 0.001$). Data in **b** and **d–g** are presented as mean values ± SEM. Statistical analysis and $P$ and $F$ values can be found in Supplementary Table 4 and original data in the Source Data file.

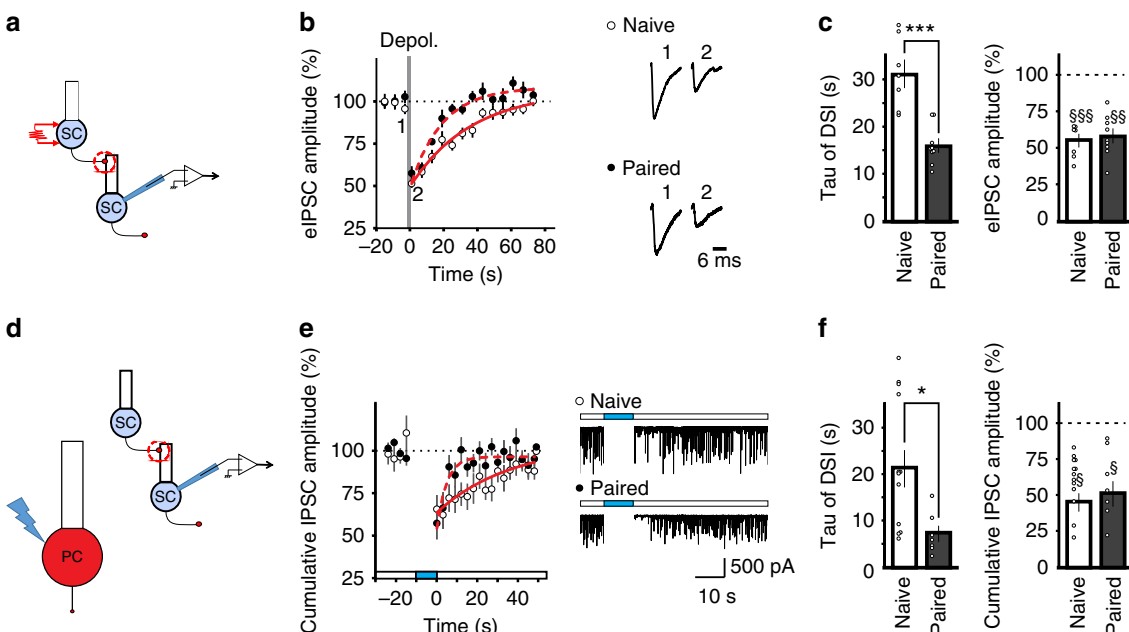

**Fig. 2 Fear conditioning accelerates the recovery of depolarization-induced suppression of inhibition (DSI). a–c** Depolarization of stellate cells induced suppression of evoked IPSCs. **a** Schematic of the experimental procedure, **b** sample traces and time course of evoked IPSCs in stellate cells from naïve ($n = 7$ cells, open circles) and conditioned (paired) mice ($n = 9$ cells, filled circles). **c** Recovery time constant of DSI and peak depression (naïve, $n = 7$ cells; paired, $n = 9$ cells). **d, e** Heterosynaptic DSI. Spontaneous IPSCs were recorded in stellate cells from L7::ChR mice and photostimulation of Purkinje cells for 10 s induced a decrease in sIPSCs. **d** Schematic of the experimental procedure, **e** examples and averaged time courses of cumulative IPSC amplitude in stellate cells from naïve mice ($n = 13$ cells) and conditioned mice ($n = 7$ cells, 15 h after fear conditioning). Best-fit exponential curves for the recovery phase: solid (naïve) and dashed (paired) red lines. **f** Recovery time constant and peak depression of heterosynaptic DSI (naïve, $n = 13$ cells; paired, $n = 7$ cells). Group comparisons of DSI recovery time in **c** and **f** were obtained with two-sided unpaired $t$-tests (*, $P < 0.05$; ***, $P < 0.001$). Effects of depolarization of MLIs and PCs on eIPSCs (**c**) and cumulative sIPSC (**f**) amplitudes, respectively, were assessed using two-sided paired $t$-tests (§, $P < 0.05$; §§, $P < 0.01$; §§§, $P < 0.001$). Data in **b**, **c**, **e**, and **f** are presented as mean values ± SEM. Statistical analysis and $P$ values can be found in Supplementary Table 4 and original data in the Source Data file.

type-specific. Because Purkinje cells also produce 2-AG and suppress GABA release from MLIs, we next tested whether fear conditioning altered depolarization-induced endocannabinoid release from Purkinje cells. Toward this end, we took the advantage of the expression of L7 in Purkinje cells and found that photostimulation of Purkinje cells in L7::ChR2 mice (Fig. 2d, Fig. S1a) induced a transient suppression of spontaneous IPSCs recorded in stellate cells (Fig. 2e, f), which was prevented by AM[251], an inverse agonist of CB1Rs (5 μM, Fig. S1b). Fear conditioning did not alter the magnitude of the peak suppression (Fig. 2e, f). Thus, endocannabinoids were produced from Purkinje cells in conditioned mice and heterosynaptically suppressed GABA release. The depression recovered with a time constant of 21.3 ± 4.0 s in naïve mice and the recovery rate was markedly accelerated after fear conditioning (7.1 ± 1.7 s, $P < 0.05$) (Fig. 2e, f). Therefore, fear conditioning enhances the degradation of endocannabinoids

that are released from stellate and Purkinje cells, suggesting that learning promotes endocannabinoid degradation.

Thus, fear conditioning does not alter the magnitude of the peak suppression following either photostimulation of Purkinje cells or depolarization of stellate cells. Because deletion or inhibition of DAGL and CB1R reduces the magnitude of the suppression, we tested whether learning also modified CB1R signaling. We applied the synthetic CB1R agonist WIN[55212-2] (5 μM) and found that miniature IPSC (mIPSC) frequency in stellate cells decreased in both naïve and conditioned mice. Overall mIPSC frequency was reduced by 43 ± 4% in naive and 48 ± 6% in conditioned mice, relative to before WIN[55212-2] application ($P > 0.05$; Fig. 3a–c). As endocannabinoid release from stellate cells in response to depolarization during DSE and DSI depends on activation of voltage-gated Ca²⁺ channels, we quantified the amplitude of Ca²⁺ currents in stellate cells upon

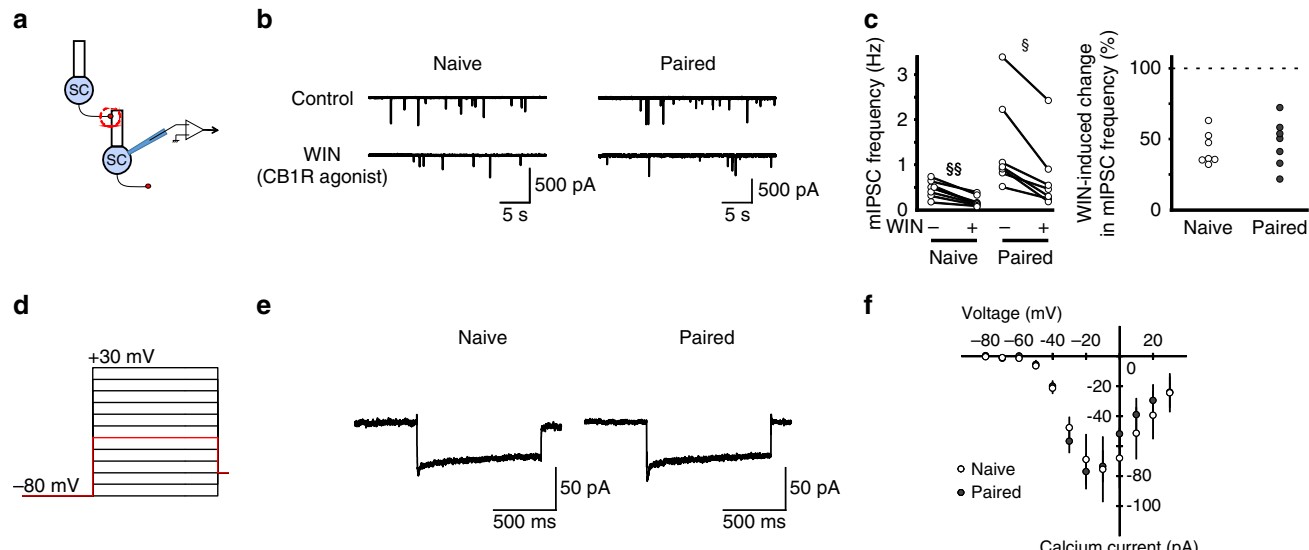

**Fig. 3 Fear conditioning does not alter CB1R signaling or calcium entry that evokes eCB release. a–c** $CB_1$ receptor signaling is not affected by fear conditioning. **a** Schematic of the experimental procedure. **b** Example traces of mIPSCs from stellate cells before and during application of CB1R agonist WIN[55212-2]. **c** WIN[55212-2] reduces mIPSC frequency in both naïve ($n = 7$ cells) and conditioned mice ($n = 7$ cells). **d, e** Fear conditioning did not change voltage-gated calcium currents in cerebellar stellate cells. **d** Voltage-clamp protocol used to evoke calcium currents. **e** Example traces of calcium currents in response to membrane depolarization to −30 mV (shown in red in **d**). For representation purpose, artifacts have been filtered out. **f** Group data showing the current–voltage relationship of calcium currents recorded in stellate cells from naïve (open circles, $n = 5$ cells) and paired (filled circles, $n = 7$ cells) animals. Effects of WIN on mIPSC frequency in naïve were compared with paired group using a two-way RM ANOVA followed by Tukey's post hoc test (§, $P < 0.05$; §§, $P < 0.01$). Data are presented as mean values ± SEM. Statistical analysis and $P$ and $F$ values can be found in Supplementary Table 4 and original data in the Source Data file.

membrane depolarization and found that this was unaltered after fear conditioning (Fig. 3d–f). Therefore, learning did not modify tonic CB1R signaling or the depolarization-evoked $Ca^{2+}$ currents that trigger endocannabinoid release from stellate cells. The magnitude of peak suppression during DSE and DSI is consistent with previous reports[4,27,31–33].

**GABA drives the accelerated degradation of endocannabinoids.**
Cerebellar stellate cells can be depolarized by stimulation of parallel and climbing fibers and co-activation of these inputs during fear conditioning should produce a strong depolarization in these neurons[34]. We therefore tested whether activation of these interneurons could *induce* an increase in endocannabinoid degradation. Since NOS promoters have been used to drive ChR2 expression in stellate/basket cells in adult mice[35], we crossed NOS-cre with floxed ChR2 mice and generated NOS::ChR2 mice and found that photostimulation activated MLIs but also Purkinje cells (Table S2). We activated ChR2 (10s-on→20s-off, 8×) to evoke GABA release in slices from naïve NOS::ChR2 mice and quantified DSE at least 2 h later (Fig. 4a, b). While the amplitude and kinetics of DSE in non-stimulated slices were similar to DSE in naïve wild type (Fig. S2a, b), prior photostimulation accelerated the recovery time of DSE from $10.9 ± 1.8$ s to $5.3 ± 0.6$ s ($P < 0.05$, Fig. 4c–g), but did not alter the amplitude of DSE (Fig. S2a). The change in recovery time of DSE after photostimulation was comparable to that observed after fear conditioning. Inhibition of MAGL prolonged the recovery rate of DSE in photostimulated stellate cells to ~15 s which is indistinguishable from the time constant quantified in non-stimulated cells in the presence of JZL (Fig. 4e–g). This indicates that the accelerated recovery rate of DSE after photostimulation of MLIs is mostly due to an increase in 2-AG degradation.

Photostimulation of MLIs causes GABA release from these inhibitory interneurons. To confirm that GABA mediates the change in recovery rate via activation of $GABA_A$-receptors, we

included $GABA_A$-R blockers (PTX and SR95531) in the bathing solution during photostimulation, and this prevented the increase in 2-AG degradation (Fig. 4d). Since GABA can trigger spontaneous glutamate release from parallel fibers[36], we applied glutamate receptor inhibitors during photostimulation. In contrast, blocking glutamate receptors did not alter the recovery time of DSE (Fig. 4g). Because photostimulation directly activated not only stellate cells, but also Purkinje cells in the NOS::ChR2 mice (Fig. S2c and Tables S1, 2), we activated Purkinje cells from naïve L7::ChR2 mice as a control and found that Purkinje cell activation did not alter the kinetics and amplitude of DSE (Fig. 4f, g). These results indicate that GABA released from inhibitory interneurons in the cerebellar cortex is required to induce an increase in endocannabinoid degradation.

It has been shown that fear conditioning enhances spontaneous GABA release onto Purkinje cells[15], and thus an increase in GABA release may contribute to the maintenance of the sustained increase in 2-AG degradation. We therefore determined whether fear conditioning induced a long-lasting increase in GABA release from cerebellar stellate cells. The amplitude of evoked inhibitory synaptic currents (eIPSCs) at stellate-to-stellate cell synapses increased after fear conditioning relative to naïve and unpaired controls using the same stimulation strength (Fig. 5a, b). The increase in eIPSC amplitude was accompanied by a decrease in the paired pulse ratio of eIPSCs (Fig. 5a, b). We also found that the frequency of mIPSCs (Fig. 5c, d and Fig. S3b) and spontaneous IPSCs (Fig. 5e, f and Fig. S3c) in stellate cells from fear conditioned mice was increased compared to naïve and unpaired controls without a change in amplitude. Therefore, fear conditioning produces a long-lasting increase in both evoked and spontaneous GABA release from stellate cells.

Since learning elevated GABA release, we next tested the prediction that $GABA_A$ receptor activity mediated enhanced 2-AG degradation after learning. Cerebellar slices from conditioned mice were incubated with $GABA_A$-R blockers, PTX (100 μM) and

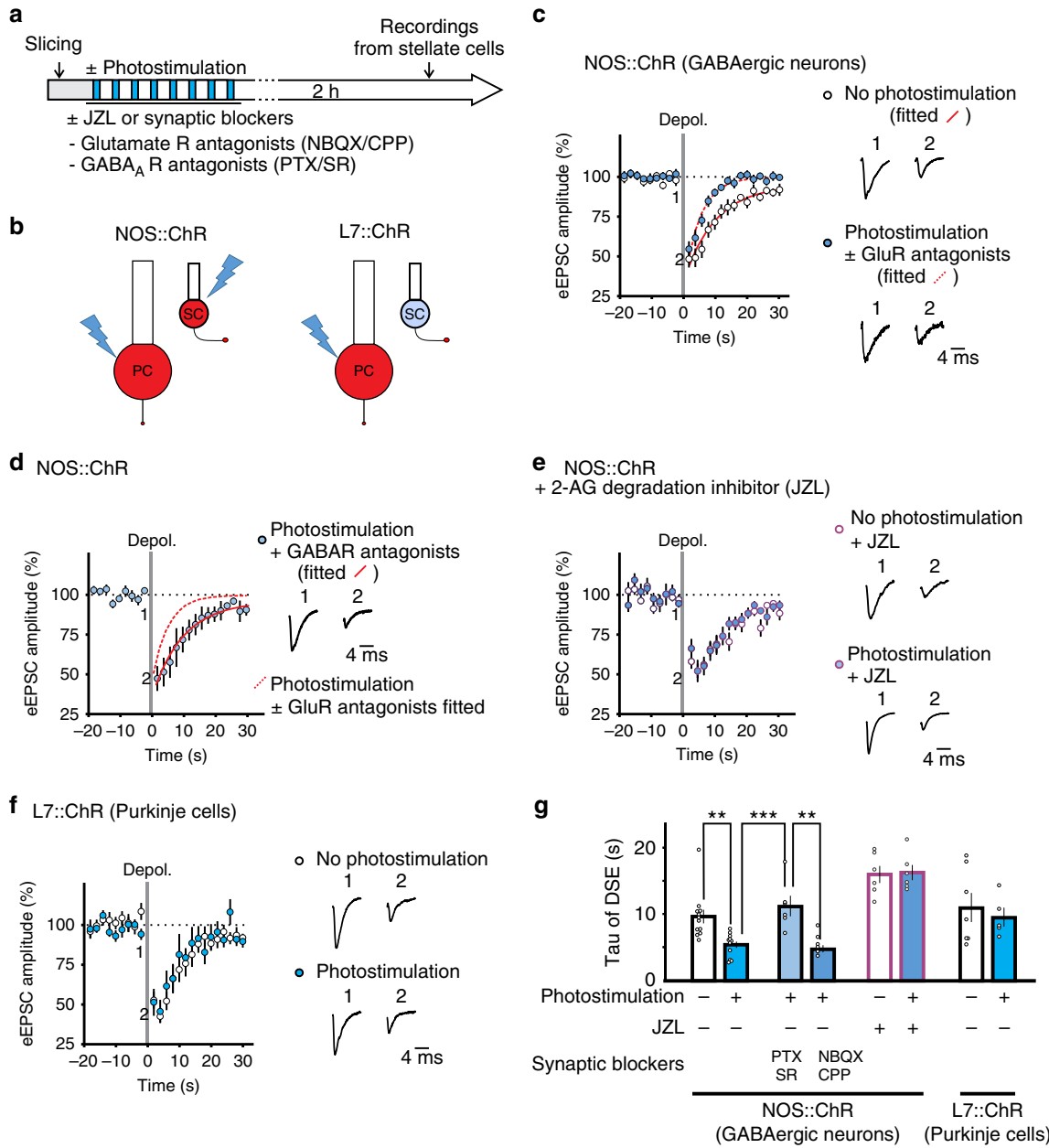

**Fig. 4 Optogenetic activation of cerebellar interneurons shortens endocannabinoid signaling. a** Cerebellar slices from NOS::ChR or L7::ChR mice were exposed to photostimulation and tested for DSE 2 h later. **b** Schematic of the cell types that express Channelrhodopsin-tomato (red) in NOS::ChR and L7::ChR mice. **c** Photostimulation (± glutamate-receptor blockers, NBQX + CPP) accelerates the recovery time of DSE relative to no stimulation control ($n = 12$ cells) in NOS::ChR mice. **d** The presence of GABA$_A$R blockers ($n = 6$ cells), during photostimulation prevented the change. **e** JZL[184], a MAGL inhibitor, increased the recovery time of DSE and abolished the difference between photostimulated ($n = 6$ cells) and non-stimulated cells ($n = 6$ cells).
**f** Photostimulation ($n = 5$ cells) failed to alter DSE in L7::ChR mice (no stimulation $n = 7$ cells). **g** Summary of DSE recovery time (photostimulated no drug, $n = 5$ cells; photostimulated + NBQX/CPP, $n = 6$ cells). Group comparisons were obtained with two-sided unpaired $t$-test (\*, $P = 0.026$; \*\*, $P = 0.006$; \*\*\*, $P = 0.007$). Data in **c–g** are presented as mean values ± SEM. Detailed statistics and exact $P$ values can be found in Supplementary Table 4 and original data in the Source Data file.

SR95531 (5 μM) for at least 3 h, and then DSE was measured to quantify the degradation rate (Fig. 5g). This treatment prolonged the recovery time of DSE from $4.2 \pm 0.6$ s (after fear conditioning) to $15.7 \pm 1.1$ s (with PTX/SR incubation, $P < 0.001$, Fig. 5h, i). Addition of the MAGL inhibitor JZL[184] after incubation with GABA$_A$-R blockers did not further increase the recovery time (Fig. 5h, i). Thus, inhibition of GABA$_A$ receptors leads to suppression of 2-AG degradation (although it is also possible that spontaneous activity of other neurons consequent to blocking GABA$_A$ receptors contributes to the change). Given that

photostimulation of MLIs and fear conditioning both increased GABA secretion and accelerated DSE recovery rate, our results suggest that a high level of MLI activity or GABA release is a physiological regulator that promotes 2-AG degradation.

**Fear conditioning selectively elevates MAGL levels in cerebellar lobule V/VI.** 2-AG is mainly metabolized by MAGL and deletion of the degrading enzyme prolongs the recovery time of DSE[4]. If learning accelerates DSE recovery rate by increasing 2-AG

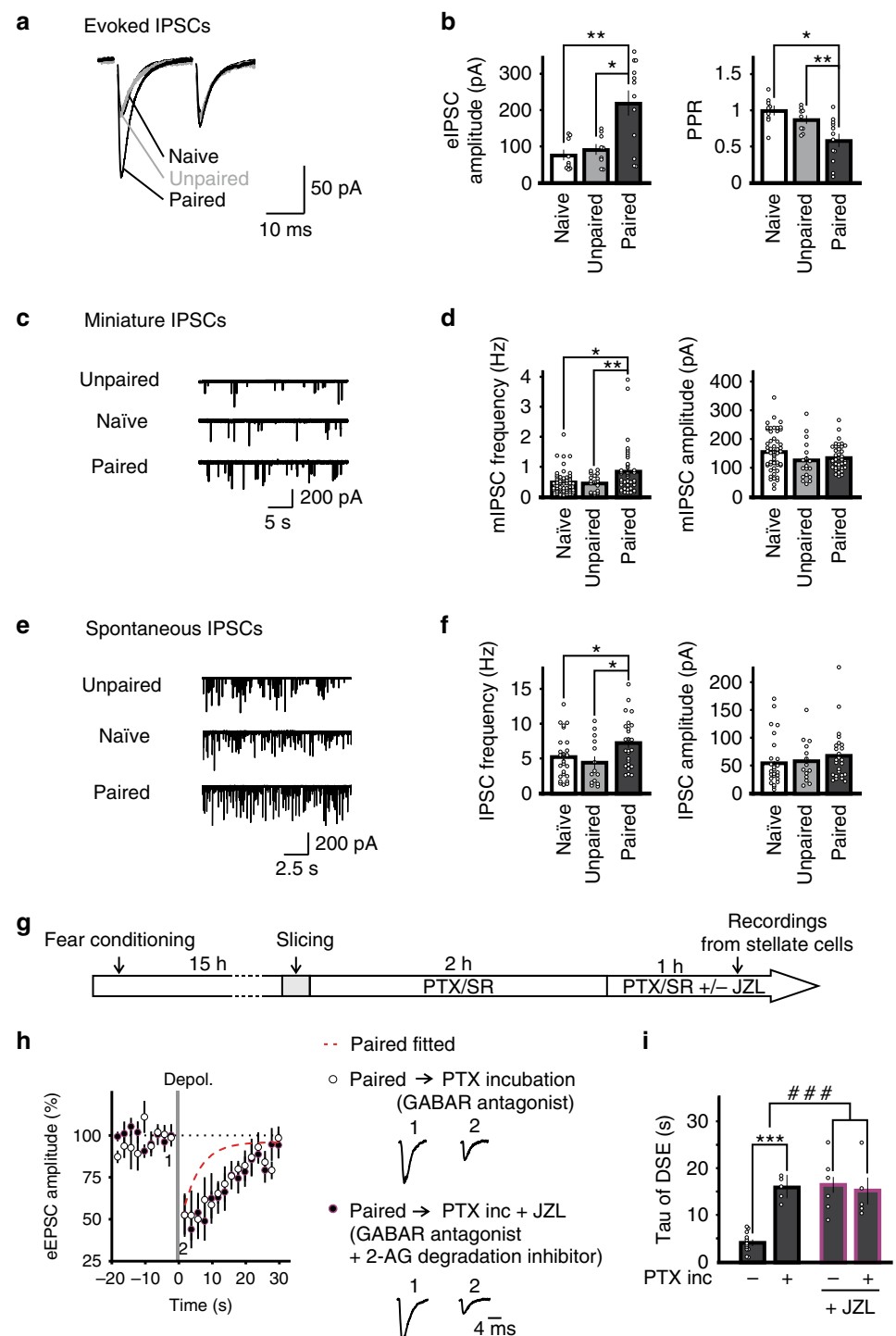

degradation, this is likely to be mediated by a change of the level of MAGL. We therefore measured the expression of MAGL protein in the cerebellar cortex using immunohistochemistry (Fig. 6a). MAGL immunoreactivity (MAGL-ir) was observed in the granule cell layer and molecular layer where the processes of Bergmann glial cells and axons of granule cells are located as described previously[4] (Fig. S4). In unpaired controls, MAGL-ir was significantly lower in lobules V/VI (25 ± 3 au), when compared to lobules IX/X (35 ± 5 au, Fig. 6b–d) which are involved in motor learning ($P < 0.05$).

After fear conditioning, the level of MAGL-ir in the molecular layer of lobule V/VI of the cerebellar cortex was increased by ~50% (37 ± 5 au; $P < 0.05$, Fig. 6), whereas the granule cells layer

was not affected ($P > 0.05$, Fig. 6). In contrast to lobules V/VI, fear conditioning did not alter MAGL-ir expression in lobule IX/X ($P > 0.05$, Fig. 6b–d). Therefore, fear conditioning selectively elevated 2-AG degrading enzyme levels in the molecular layer of lobules V/VI.

**Fear conditioning reduces tonic endocannabinoid signaling in lobules V/VI.** Acceleration of 2-AG degradation following fear conditioning may reduce tonic 2-AG levels in the cerebellum. We tested this possibility using two approaches.

First, we directly measured 2-AG levels in cerebellar lobules I to VI using LC-MS. We found that the 2-AG content was

**Fig. 5 Fear conditioning elevates GABA release and blocking GABA$_A$ receptors reverses learning-induced acceleration of 2-AG degradation. a–f** Fear conditioning increases GABA release. **a, b** IPSCs were evoked by two consecutive stimuli and recorded in a stellate cell from naïve ($n = 9$ cells), unpaired controls ($n = 9$ cells), and conditioned mice ($n = 12$ cells). **a** Representative traces. **b** The amplitude of the first eIPSC and paired-pulse ratio (PPR = IPSC$_{2nd}$/IPSC$_{1st}$). Two-sided unpaired $t$-test, eIPSC amplitude: *, $P = 0.006$; **, $P = 0.003$; PPR: *, $P = 0.002$; **, $P = 0.019$. **c–f** mIPSCs (**c, d**) and spontaneous IPSCs (**e, f**) were recorded in cerebellar stellate cells. **c** Representative mIPSC traces. **d** Summary of mIPSC frequency (two-sided unpaired $t$-test *, $P = 0.006$; **, $P = 0.046$) and amplitude (naïve $n = 54$ cells; unpaired $n = 19$ cells; paired $n = 39$ cells). **e** Representative spontaneous IPSC traces. **f** Summary of sIPSC frequency (two-sided unpaired $t$-test, *, $P = 0.035$; **, $P = 0.019$) and amplitude (naïve $n = 27$ cells; unpaired $n = 14$ cells; paired $n = 25$ cells). **g–i** Inhibition of GABA$_A$ receptor reverses learning-induced change in 2-AG degradation. **g** Cerebellar slices from conditioned mice were incubated in 100 μM PTX and 5 μM SR-95531 ($n = 5$ cells) or with the addition of JZL[184] ($n = 5$ cells) for 3 h before quantifying DSE (FC + JZL[184] without GABA$_A$R blockers $n = 6$ cells). **h** Sample traces and averaged time course of DSE after each treatment (PTX + SR-95531, $n = 5$ cells; PTX + SR-95531 + JZL[184], $n = 5$ cells) compared with fitted time course without inhibitor treatment (dashed line). **i** Summary of the recovery time of DSE (no inhibitor, $n = 5$ cells; JZL[184], $n = 6$ cells; PTX + SR-95531, $n = 5$ cells; PTX + SR-95531 + JZL[184], $n = 5$ cells). The effect of JZL on DSI was assessed by a two-way ANOVA (#, $P < 0.001$) followed by Tukey's post hoc test (***, $P < 0.001$). Data in **b, d, f, h** and **i** are presented as mean values ± SEM. Statistical analysis and $P$ and $F$ values can be found in Supplementary Table 4 and original data in the Source Data file.

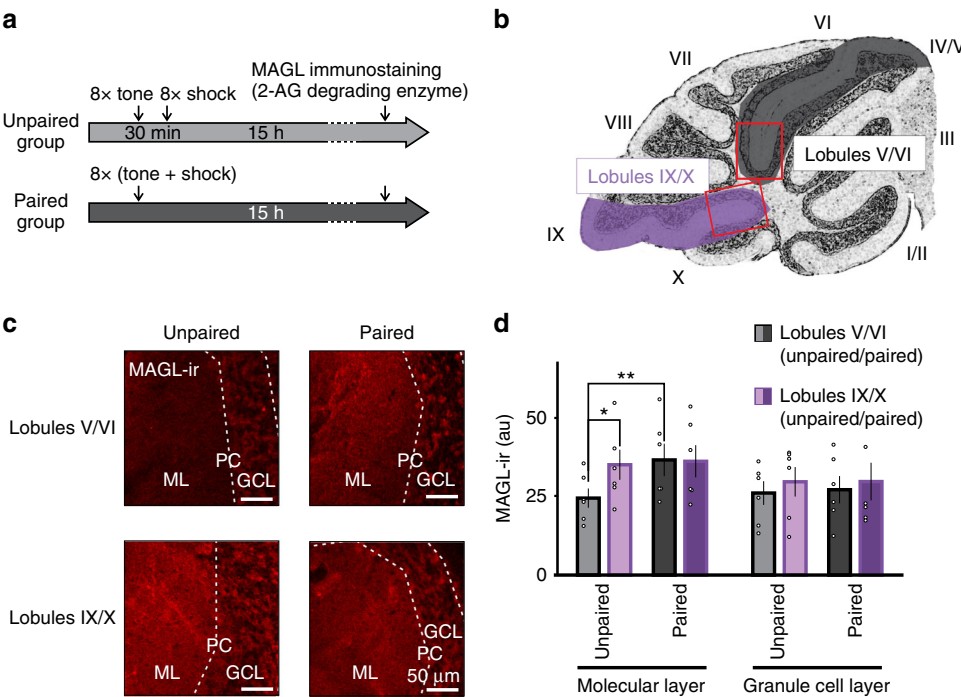

**Fig. 6 Fear conditioning increases the expression of monoacylglycerol lipase (MAGL), a 2-AG degrading enzyme, in lobules V/VI of the cerebellar vermis. a** Experimental protocol. **b** Schematic illustration of a sagittal section of the cerebellar vermis and regions where images were taken (red boxes). **c** Representative immunofluorescence images of MAGL immunostaining in lobules V/VI and lobules IX/X of the cerebellum from unpaired and fear conditioned mice. **d** Group data showing a higher level of MAGL immunoreactivity (MAGL-ir) in the molecular layer of lobules V/VI compared to lobules IX/X in unpaired control animals ($n = 6$ animals; ≥2 slices/animal). After fear conditioning ($n = 6$ animals), MAGL expression is increased in lobules V/VI compared to unpaired controls. MAGL-ir levels in the granule cell layer of lobules V/VI is similar to lobules IX/X and is not affected by fear conditioning. Group comparisons were performed with a two-way RM ANOVA ($P = 0.021$) followed by Tukey's post hoc test (*, $P = 0.008$; **, $P = 0.003$). Data are presented as mean values ± SEM. Statistical analysis and $P$ and $F$ values can be found in Supplementary Table 4 and original data in the Source Data file.

significantly reduced in conditioned mice ($6.8 \pm 1.8$ ng/mg of tissue, Fig. 7a and Methods) relative to naïve ($9.3 \pm 2.4$ ng/mg of tissue, $P < 0.05$, Fig. 7a) and unpaired controls ($8.2 \pm 2.9$ ng/mg of tissue, $P < 0.05$, Fig. 7a). Therefore, there is an overall reduction in the 2-AG level in lobule V/VI, the site at which learning increases the degradation of 2-AG.

Second, we used a functional assay to quantify the tonic endocannabinoid levels. Endocannabinoid signaling which is mediated via CB1Rs in stellate cells is tonically active and leads to a reduction in GABA release[37]. Bath application of the neutral CB1R antagonist NESS[0327] (0.5 μM) increased mIPSC frequency in stellate cells from lobule V/VI of naïve animals (Fig. 7b–d), indicating the presence of an endocannabinoid tone. Application of inverse agonists AM[251] and AM[281] (5 μM) that inhibit both the

constitutive and agonist-evoked CB1R activity also enhanced mIPSC frequency to a level that was comparable to the increase induced by NESS[0327] (Fig. 7d and Fig. S5a–b). These results suggest the presence of tonic endocannabinoid signaling, rather than constitutive CB1R activity in naïve mice. Because MAGL-ir is higher in vermal lobules IX/X than lobules V/VI (Fig. 6), this may reduce 2-AG tonic levels to a greater extent in lobules IX/X. Consistent with the prediction, NESS[0327] failed to increase mIPSC frequency in stellate cells from vermal lobules IX/X (Fig. 7d and Fig. S5c). Therefore, the endogenous cannabinoid tone is selectively present in cerebellar lobules V/VI.

We next quantified tonic eCB signaling in lobules V/VI after fear conditioning. In contrast to naïve and unpaired controls, NESS[0327] now failed to alter mIPSC frequency and CB1R

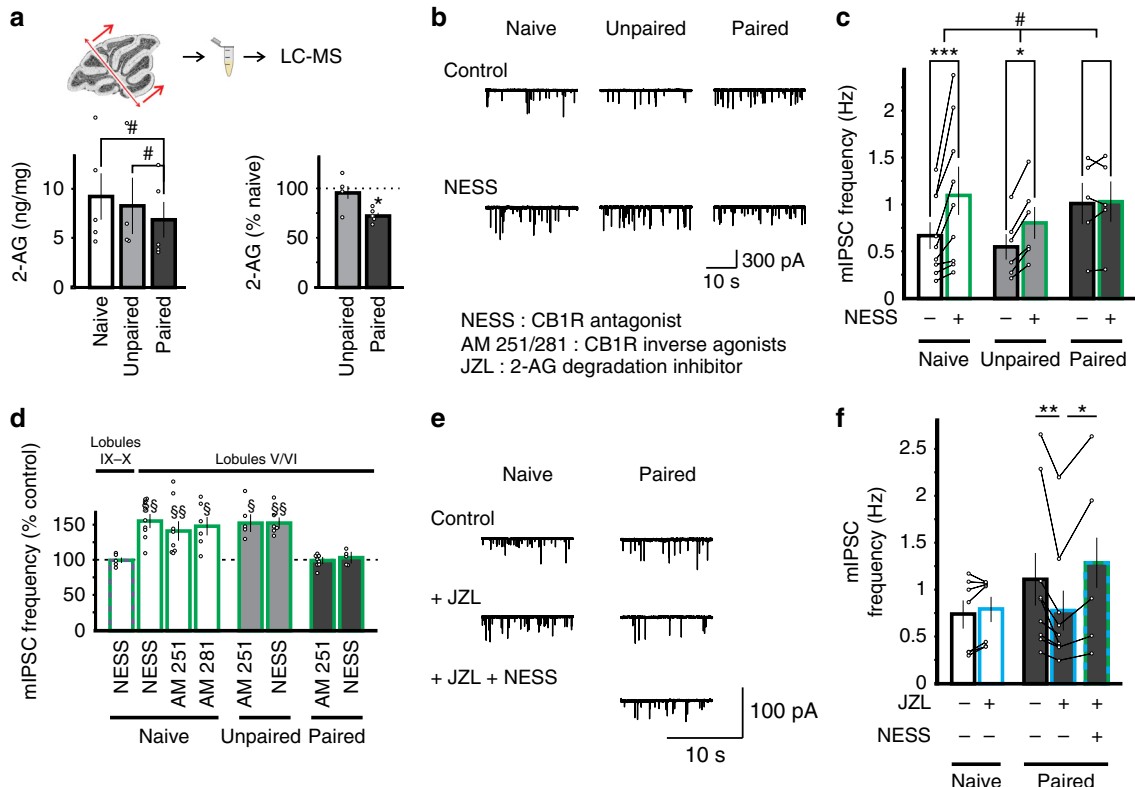

**Fig. 7 Increased 2-AG degradation after fear conditioning abolishes the tonic endocannabinoid signaling in lobules V/VI. a** Top, lobules I to VI/VII of the cerebellar vermis were dissected out and lipids were extracted for LC-MS measurements. Bottom, The 2-AG content from conditioned mice (paired, $n = 5$ animals, 1 sample/animal) was significantly lower when compared to naïve mice ($n = 5$ animals) or after the unpaired protocol ($n = 4$ animals; one-way ANOVA, $P = 0.013$; tukey post-hoc, paired vs naïve, $P = 0.014$; paired vs unpaired, $P = 0.045$). **b** Representative mIPSC recordings from stellate cells in lobules V/VI of naïve and unpaired controls and conditioned mice before and during application of a CB1R antagonist, 0.5 μM NESS[0327]. **c** Group data showing that NESS application elevated mIPSC frequency in naïve and unpaired, but not paired animals (two-way RM ANOVA, $P = 0.034$, #; followed by Tukey's post hoc test: *, $P = 0.026$; ***, $P < 0.001$). **d** Effects of a CB1R neutral antagonist NESS[0327] (naïve lobules IX/X $n = 5$ cells, naïve lobules V/VI $n = 9$ cells, unpaired $n = 6$ cells, paired $n = 5$ cells), inverse agonists, 5 μM AM[251] (naïve $n = 8$ cells, unpaired $n = 5$ cells, paired $n = 8$ cells) and 5 μM AM[281] ($n = 6$ cells), on mIPSC frequency in cells from lobules V/VI and IX/X were normalized to before drug application. **e** Representative mIPSC recordings in stellate cells before and during application of the MAGL inhibitor JZL[184] in slices from naive and fear conditioned mice. Application of NESS[0327] to inhibit CB1Rs restored mIPSC frequency in conditioned animals to control levels. **f** Group data showing that JZL[184] or JZL[195] ($n = 7$ cells) did not change mIPSC frequency in control mice (two-sided paired t-test, $P = 0.142$). In contrast, JZL[184] and JZL[195] ($n = 9$ cells) reduced mIPSC frequency in paired group (two-sided paired t-test, $P = 0.008$), an effect reversed with addition of the CB1R antagonist NESS[0327] ($n = 5$ cells, two-sided paired t-test, $P = 0.039$). No difference between the effect of JZL[184] and JZL[195] application was observed and these data were pooled. Data in **a**, **c**, **d**, and **f** are presented as mean values ± SEM. Statistical analysis and $P$ and $F$ values can be found in Supplementary Table 4 and original data in the Source Data file.

inverse agonists did not increase spontaneous GABA release in stellate cells, suggesting that learning had abolished tonic eCB signaling (Fig. 7b–d). This was not due to an impairment in CB1R signaling because the CB1R agonist WIN[55212-2] suppressed mIPSC frequency in conditioned mice (Fig. 3c). Both NESS[0327] and AM[251] increased mIPSC frequency, enhancing spontaneous GABA release in slices from unpaired control mice. Therefore, associative learning suppressed tonic eCB signaling, while CB1R signaling remained intact. Because learning elevated degrading enzyme MAGL levels (Fig. 6), we tested whether MAGL activity could lower the tonic 2-AG level in conditioned mice. We found that application of JZL[184] in cerebellar slices from conditioned mice reduced mIPSC frequency and this was reversed by a CB1R antagonist (Fig. 7e, f). Therefore, elevation of MAGL can lead to a reduction in tonic 2-AG signaling. Endocannabinoids also suppress action potential firing in MLIs[38]. Application of AM[251] increased action potential frequency in stellate cells from naïve animals in the presence of GABA$_A$-R blockers, but failed to alter interneuron spike activity in conditioned mice (Fig. S5d). Together these results

indicate that the endogenous 2-AG levels and cannabinoid signaling were reduced after fear conditioning.

A decrease in tonic 2-AG levels is predicted to dis-inhibit, and thus enhance spontaneous GABA release. Indeed, fear conditioning increased the frequency of mIPSCs (Fig. 5d) in stellate cells, compared to naïve and unpaired controls. Application of the synthetic CB1R agonist WIN[55212-2] produced a greater reduction in mIPSC frequency in stellate cells from conditioned mice ($0.66 \pm 0.15$ Hz) compared to naïve mice ($0.25 \pm 0.04$ Hz; $P = 0.01$; Fig. 3a–c), consistent with a lower level of tonic eCB after fear conditioning. Together our results show that learning accelerated the recovery rate of DSE and DSI, elevated MAGL expression, and reduced tonic 2-AG levels, providing strong evidence for an activity-dependent increase in 2-AG degradation.

**Activation of hM3Dq in Purkinje cells impairs fear memory consolidation by enhancing eCB signaling.** Our data so far suggested that fear conditioning increased GABA release, and

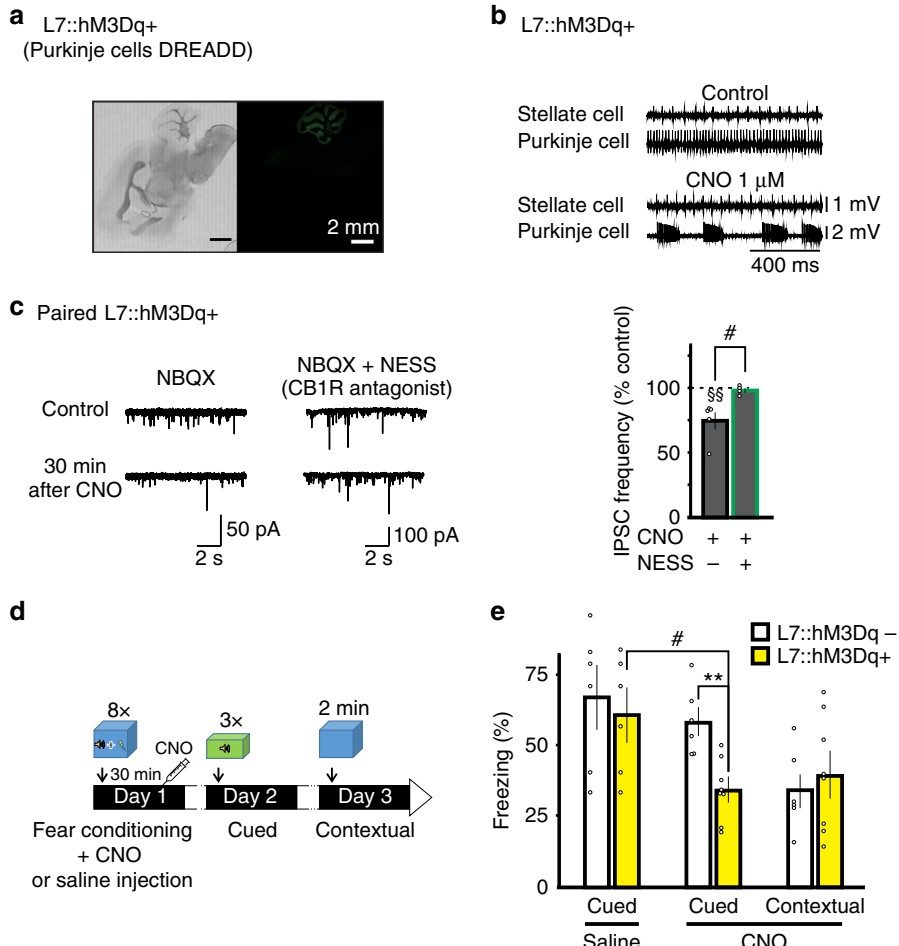

**Fig. 8 Pharmacogenetic activation of Gq pathways in Purkinje cells in vivo after fear conditioning disrupts cued fear memory retention. a** Bright field and fluorescence images of mCitrine from an L7::hM3Dq+ mouse. **b** Paired extracellular recordings from a stellate and Purkinje cell before (top) and during (bottom) application of CNO. **c** Left, representative traces of IPSCs recorded from stellate cells before (top) and 30 min after (bottom) CNO application (*n* = 5 cells). Center, NESS0327 was present throughout the experiment (*n* = 4 cells). Right, effects of CNO on IPSC frequency in stellate cells normalized to control. Group comparisons by two-way RM ANOVA (#, *P* = 0.016) followed by Tukey's post hoc test (§§, *P* = 0.002). **d** Protocol used for fear conditioning and memory retention testing. **e** Effects of DREADD activation on cued and contextual fear memories (CNO: L7::hM3Dq+ *n* = 8 animals, L7::hM3Dq− *n* = 6 animals; Saline: L7::hM3Dq+ *n* = 6 animals, L7::hM3Dq− *n* = 6 animals). Group comparisons by two-sided paired *t*-test (#, *P* = 0.024; **, *P* = 0.001). Data in **c** and **e** are presented as mean values ± SEM. The representative experiments in **a** and **b** have been independently replicated more than three times (see Tables S1 and S2). Statistical analysis and *P* and *F* values can be found in Supplementary Table 4 and original data in the Source Data file.

thereby accelerated 2-AG degradation and reduced tonic eCB signaling. These changes might be important for the consolidation of associative memories. Activation of the Gq/PLCβ pathway in cerebellar Purkinje cells is known to induce endocannabinoid release[39,40]. We next investigated whether the selective activation of Gq DREADD receptors in cerebellar Purkinje cells after learning, impaired memory retention[39,40].

We expressed hM3Dq, a Gq-coupled DREADD in cerebellar PCs using the L7 promoter (Fig. 8a and Tables S1, 2). Application of clozapine N-oxide (CNO) (1 μM) induced a transient change in the firing pattern in PCs (Fig. 8b) and reduced mIPSC frequency recorded in stellate cells in slices from conditioned mice. The latter was prevented by the presence of NESS0327 (Fig. 8c), suggesting that endocannabinoids were released following the activation of hM3Dq in PCs.

Mice were subjected to a fear conditioning paradigm in context A (Fig. 8d). Both L7::hM3Dq+ and L7::hM3Dq− mice exhibited a similar high level of exploratory behavior during the 2-min habituation period and a similar level of tone-evoked freezing during acquisition (Table S3). Mice were then injected with CNO (0.5 mg/kg) to activate hM3Dq, 30 min after acquisition and were

probed for cued memory retention by determining their response to tone alone in a novel context the next day (Fig. 8d). While both genotypes exhibited virtually no freezing prior to the first tone, the average tone-evoked freezing was significantly reduced in hM3Dq+ animals (34 ± 4%) compared to hM3Dq− (58 ± 5%, *P* < 0.01, Fig. 8e and Table S3). Contextual memory retention was tested the following day (Fig. 8d). Mice were exposed to context A (without tone) and both genotypes exhibited a similar level of freezing (Fig. 8e and Table S3). Therefore cued, but not contextual, fear memory retention was impaired after activation of hM3Dq receptors in Purkinje cells. As a control, hM3Dq+ mice receiving saline injection exhibited a level of freezing to tones, which was comparable to that in hM3Dq− animals (Fig. 8e and Table S3), suggesting that expression of hM3Dq+ itself did not disrupt memory consolidation. Since CNO is converted to clozapine in vivo[41,42], administration of CNO in L7::hM3Dq− mice controlled for any off-target effects of clozapine, and this treatment did not alter memory retention compared to saline injection (*P* > 0.05, Fig. 8e and Table S3). While CNO reduced locomotor activity when measured at 30 min after CNO injection, this effect was absent 24 h later, at the time of the memory

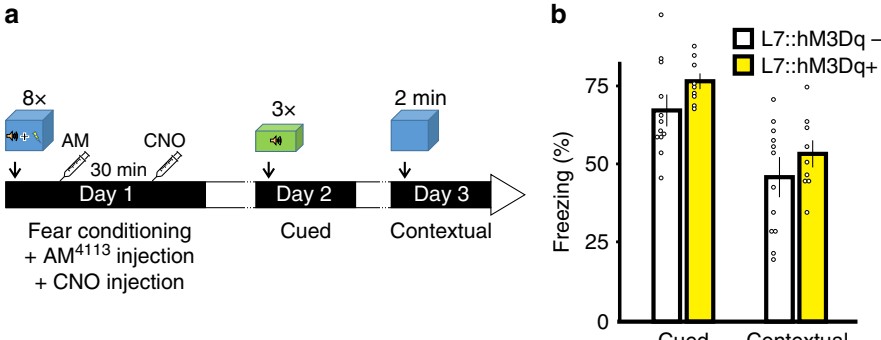

**Fig. 9 Endocannabinoid receptor antagonist prevents the disruption of memory consolidation by activation of Gq pathways in Purkinje cells. a** The protocol used for fear conditioning and memory retention testing was identical to the one described in Fig. 8, except that both L7::hM3Dq+ ($n = 9$ animals) and L7::hM3Dq− ($n = 12$ animals) mice received an additional i.p. injection of AM[4113] (3 mg/kg), a CB1 receptor neutral antagonist immediately after acquisition, 30 min before the CNO injection. **b** Group data showing that activation of Gq DREADD in Purkinje cells no longer impaired cued memory retention following injection of AM[4113]. Group comparisons by two-sided paired *t*-test (L7::hM3Dq+ (cued) CNO vs CNO + AM, $P < 0.001$). Data in **b** are presented as mean values ± SEM. Statistical analysis and *P* values can be found in Supplementary Table 4 and original data in the Source Data file.

retention test (Fig. S6). Thus, reduced freezing during the memory retention test in hM3Dq+ mice is unlikely to be due to a residual locomotor effect of CNO. Our results show that chemogenetic stimulation of Gq pathways in PCs disrupted memory consolidation.

Because activation of Gq pathways in PCs causes eCB release, learning-induced suppression of eCB signaling in the cerebellum is likely to play a crucial role in the consolidation of fear memories. To this end, we first tested whether CB1Rs mediate the Gq-dependent disruption of memory consolidation. Both L7::hM3Dq+ and L7::hM3Dq− mice were subject to the fear conditioning paradigm and received an i.p. injection of AM[4113] (3 mg/kg), a CB1R neutral antagonist[36] immediately after acquisition, and this was followed by a CNO injection 30 min later (Fig. 9a). Mice were tested for cued memory retention next day. We found that administration of AM[4113] did not alter memory retention in L7::hM3Dq− mice, but prevented the disruption of memory formation induced by CNO injection in L7::hM3Dq+ animals (Fig. 9b and Table S3). Therefore, enhanced eCB signaling via activation of a Gq pathway in PCs is responsible for the attenuation of memory consolidation.

Second, considering that fear conditioning suppressed 2-AG signaling by enhancing its degradation, we tested whether activation of the Gq pathway in PCs disrupted the learning-induced increase in MAGL expression. CNO was administered to L7::hM3Dq+ and hM3Dq− mice 30 min after the fear conditioning paradigm, and MAGL-ir was quantified next day (Fig. 10a). We found that L7::hM3Dq+ mice had a lower level of MAGL-ir in the molecular layer, relative to L7::hM3Dq− mice (Fig. 10b, c). A decrease in 2-AG degradation is predicted to increase tonic eCB levels and suppress GABA release. Indeed L7::hM3Dq+ mice exhibited a lower mIPSC frequency than L7::hM3Dq− mice (Fig. 10d, e). The application of a CB1R blocker enhanced the mIPSC frequency in L7::hM3Dq+ mice, suggesting an elevated tonic eCB level (Fig. 10d, e). Therefore, activation of Gq pathways in PCs not only attenuated memory consolidation, but also prevented the learning-induced suppression in eCB signaling, when quantified 24 h later. Together these results suggest that learning-induced suppression of eCB signaling in the cerebellum is required for memory consolidation.

## Discussion

It is well established that degradation of neuromodulators, such as endocannabinoids, controls the temporal profile of their

modulatory action, and the loss of function of MAGL alters behaviors, such as learning and memory, and antinociceptive effects[23]. However, whether there is physiological regulation of the 2-AG degradation rate is not known. Our results demonstrate that endocannabinoid degradation is dynamically regulated by neuronal activity and actively participates in learning and memory. We find that learning enhances GABA release and thereby promotes 2-AG degradation. This requires an increase in MAGL expression and leads to a reduction in tonic 2-AG levels. At a behavioral level, activation of Gq-DREADD in Purkinje cells evoked the release of endocannabinoids and impairs the consolidation of fear memory via the activation of CB1Rs. These results demonstrate that neuronal activity can regulate the expression of MAGL and thereby the degradation of 2-AG. This form of plasticity occurs in vivo after learning and is responsible for memory consolidation.

The therapeutic potential of regulating the rate of degradation has been demonstrated using cholinesterase inhibitors for the treatment of cognitive deficits[1]. Inhibition of endocannabinoid degradation by MAGL can reduce anxiety-like behaviors in rodents, but also alters learning and memory and produces antinociceptive effects[6–8]. This is due to the enhancement of 2-AG signaling in multiple neural circuits. A physiological regulation of 2-AG degradation rate would therefore be expected to occur within the activated circuit and selectively alter the relevant behavior. Although degrading enzymes for several major neuromodulators have been well characterized and their role in behavior clearly defined, few studies have examined how an experience, such as stress or alcohol abuse, regulates the expression of these enzymes[9,43–45]. Our study shows that associative fear conditioning accelerated 2-AG degradation, elevated MAGL expression level and reduced tonic 2-AG levels. Therefore, there is a physiological regulation of 2-AG degradation rate that shortens the temporal window of 2-AG action and lowers tonic 2-AG activity.

Our study also addresses the fundamental question of whether neuronal activity can regulate the rate at which a neuromodulator is degraded. We show that a brief photo-stimulation of GABAergic interneurons is sufficient to induce a lasting increase in 2-AG degradation in the cerebellum. This change is due to the activation of GABA$_A$ receptors, but not glutamate receptors. Thus, the activity of inhibitory interneurons is a major regulator of 2-AG degradation. Further, such interneuron activity-dependent acceleration of 2-AG degradation occurs in vivo after learning. Sensory stimulation/deprivation and drugs of

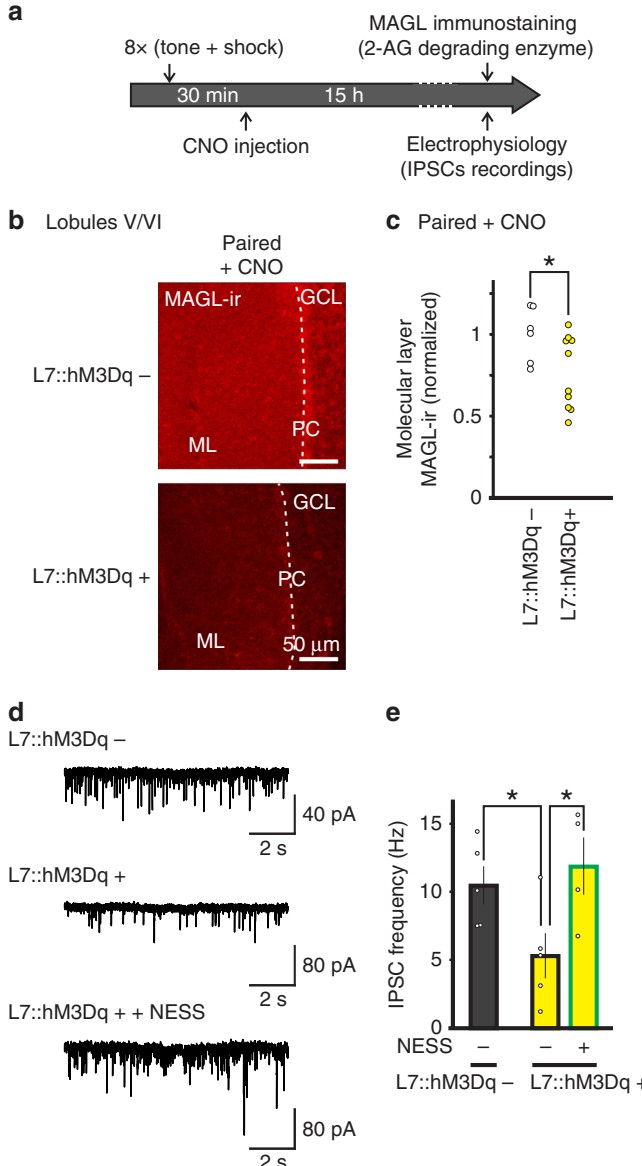

**Fig. 10 Activation of Gq pathways in Purkinje cells after learning reduces MAGL expression in the cerebellum and blocks the learning-induced increase in GABA release. a** Experimental protocol. Animals were subject to fear conditioning on day 1 and administered with CNO 30 min later, and immunostaining of MAGL and sIPSC recordings were performed 15 h after learning. **b** Typical MAGL immunofluorescence images obtained from L7::hM3Dq+ and L7::hM3Dq− mice (see below for the number of independent replications). **c** Group data showing a lower MAGL immunoreactivity in L7::hM3Dq+ ($n = 10$ animals with measurement in at least two different slices each) compared to L7::hM3Dq− mice ($n = 6$ animals with measurement in at least two different slices each, two-sided paired *t*-test, $P = 0.044$). **d** Representative traces of spontaneous IPSCs recorded in stellate cells from L7::hM3Dq− (top) and L7::hM3Dq+ (middle and bottom). **e** Group data showing that sIPSC frequency in stellate cells from L7::hM3Dq+ mice ($n = 5$ cells) is lower than in stellate cells from L7::hM3Dq− mice ($n = 5$ cells, two-sided unpaired *t*-test, $P = 0.044$) and increased during application of the CB1 receptor antagonist NESS0327 ($n = 4$ cells, two-sided unpaired *t*-test, $P = 0.041$). Data in **e** are presented as mean values ± SEM. Statistical analysis and *P* values can be found in Supplementary Table 4 and original data in the Source Data file.

abuse also alter GABA release in multiple brain regions[46–49], and so this may be a widespread mechanism that controls an experience-dependent regulation of endocannabinoid tone. Our findings identify a previously unappreciated reciprocal interaction between two major transmitters, in which GABA release drives endocannabinoid degradation.

This form of neural plasticity exhibits two important features. First, since endocannabinoids suppress neurotransmitter release, a reduction in 2-AG tone in turn elevates GABA release by disinhibition[50]. This reciprocal interaction leads to a sustained elevation of GABA release and a lasting reduction in endocannabinoid levels via a self-sustained positive feedback loop, manifesting a form of learning-induced metaplasticity. Within the cerebellum, learning elevated the activity of inhibitory neurons (the present study) and excitatory input onto Purkinje cells[16], providing a mechanism for the synaptic consolidation of fear memory. Second, in contrast to input-specific synaptic plasticity, a change in endocannabinoid degradation alters both excitatory and inhibitory synaptic transmission in a synapse-independent manner[4,38], as well as neuronal excitability of inhibitory interneurons, consequently modifying the activity of the entire cerebellar circuit.

While the cerebellum has traditionally been considered as controlling balance and motor coordination, clinical studies show that the cerebellum is also responsible for cognitive and emotional processing. There is strong evidence supporting a cerebellar non-motor role in the consolidation of fear memory, social behavior, and autism[11,51,52]. This could result from information processing within the cerebellum and its extensive connections with cortical and sub-cortical regions[53]. Pavlovian fear conditioning is used as a model for emotional learning and memory. Although the primary role of the amygdala in associative fear learning is well established[54], our results demonstrate that a transient activation of Gq pathways in cerebellar Purkinje cells after fear learning is sufficient to impair memory consolidation. This is consistent with the finding that reversible inhibition of cerebellar activity with TTX disrupts memory consolidation[11] and protein synthesis inhibitors prevent reconsolidation of fear memories[55]. These approaches allowed us to disrupt consolidation without affecting learning and so avoided the complications that can be present when using knockout mice[56]. Furthermore, inactivation of the cerebellar vermis after memory acquisition disrupts fear memories, assessed by conditioned freezing, bradycardia, and inhibitory avoidance tasks in animals and humans[12,13,15,57–62]. Therefore, the cerebellum is also critical for the consolidation of associative fear memory. Activation of Gq pathways increased endocannabinoid levels and reduced GABA release in the cerebellum, and reversed the learning-induced change in the cerebellum. This result reveals the neuronal mechanisms underlying emotional learning, an underappreciated, but clinically important, non-motor function of the cerebellum.

Endocannabinoids are critically involved in several aspects of emotional memory processing. Overwhelming evidence indicates that endocannabinoid signaling is essential for the extinction of fear memories and also can impair memory retrieval[7]. In contrast, the effects of endocannabinoids on memory consolidation are less clear[19,20,63–65]. Systemic administration of CB1R agonists after learning impairs memory consolidation and CB1R antagonists improves it, consistent with idea that reduced endocannabinoid signaling facilitates memory consolidation[19,20]. However, local application of CB1R agonists and antagonists into the hippocampus, amygdala, and cerebellum produce conflicting results[10,65]. Our finding that fear conditioning enhanced endocannabinoid degradation provides evidence for a learning-induced

downregulation of 2-AG signaling. Given that endocannabinoid signaling impairs memory retrieval and promotes extinction, a reduction in 2-AG signaling is expected to facilitate the formation of fear memory. Indeed, we show that the consolidation of fear memory was disrupted by selective activation of Gq-DREADD in Purkinje cells, which evoked release of 2-AG, reduced the level of the degrading enzyme, and elevated tonic 2-AG levels. The deficit in memory consolidation was also prevented by administration of a CB1R antagonist. Therefore, a learning-induced reduction in 2-AG signaling in the cerebellum is critical for memory consolidation. This is consistent with the observation that CB1R agonists impair the consolidation of fear memory while antagonists do the reverse[19,20] and may be a common pathway that is needed for the consolidation of associative fear memories. Activity-dependent neuromodulator degradation is a previously unrecognized mechanism for synaptic plasticity which is normally described in terms of changes in receptor signaling and the release of neuro-modulators[66–68].

We find that endocannabinoid signaling not only changes with experience, but also appears to be lobule specific, as the expression level of MAGL in naïve animals in the molecular layer of lobule IX/X is elevated relative to that of lobule V/VI (Fig. 6). We also found a higher mIPSC frequency (Fig. S5c) and a lack of effect of the CB1 receptor antagonist on mIPSC frequency in lobule IX/X (Fig. 7d), suggesting a lower level of endocannabinoid tone in these lobules. Altogether these results suggest that MAGL is constitutively upregulated in the molecular layer of lobule IX/X, but the underlying mechanism remains to be determined. This may explain the discrepancy in tonic endocannabinoid levels in the cerebellar cortex with application of CB1 receptor antagonists increasing spontaneous GABA release in some studies[33], but not others[32]. Considering the neural basis for the diverse functions of the cerebellum lies in its structural compartmentalization (e.g. vermal lobule V/VI for emotional memory formation; lobule IX/X for motor learning), experience-dependent plasticity is also expected to be lobule specific. Indeed, we and others show that associative fear learning induces long-term potentiation (LTP) at both excitatory and inhibitory synapses and promotes MAGL expression in lobule V/VI, not in lobule IX/X. In contrast, motor learning selectively enhances feed-forward inhibitory connectivity in lobule IX/X while fear conditioning increases inhibition growth in lobule V/VI[14].

Neuromodulator signaling shapes nervous system *function* and behavior by modulating synaptic transmission and the intrinsic excitability of neurons. The enzymatic degradation of these compounds controls the temporal profile of their modulatory action. Given that the therapeutic potential of regulating the rate of degradation has been explored in both clinical and pre-clinical studies[69–71], physiological regulation of this process by neuronal activity is a previously unappreciated mechanism for modulating behavior. This form of plasticity is expected to have a selective impact on the activity of active circuits and could provide an effective way to alter behavior.

## Methods

**Animals**. Animals for this study were initially purchased from the Jackson laboratory (Bar Harbor, ME) and breeding colonies were subsequently maintained in our animal facility (C57Bl/6J wild-type stock 000664). All the mice were on a C57Bl/6J background. Two cre mouse lines, L7::CRE (B6.129-Tg(Pcp2-cre)2Mpin/J) (stock 004146) and NOS::CRE (B6.129-Nos1<tm1(cre) Mgmj>/J) (Stock 017526), were crossed with floxed ChR (B6.Cg-Gt(ROSA)26Sortm27.1(CAG-COP4*H134R/tdTomato) Hze/J) (stock 012567) and floxed Gq-DREADD (Gt (ROSA)26Sortm2 (CAG-CHRM3*,-mCitrine)Ute/J) (stock 026220) to generate L7::ChR, NOS::ChR, and L7::hM3Dq mutant mice. To assess the expression and function of ChR and hM3Dq[72], tdTomato and mCitrine fluorescence intensity was examined in brain sections prepared from double mutant mice (Table S1) and the effects of photostimulation and application of CNO on action potential firing in cerebellar neurons were quantified (Table S2).

Only P18 to P110 male mice were used in this study. While most of the mice used in electrophysiology experiments were P18 to P50, experiments quantifying DSE, PPR, and effects of CB1R blockers on mIPSCs have been replicated in older mice (up to P110) to match the behavioral experiments. Since results were indistinguishable from those obtained using P18-50 mice, data were pooled.

Breeding colonies were maintained in our animal facility on a 12 h light/dark cycle, with ad libitum food and water supply. Animals were never single housed and all precautions were taken to avoid any stressful environment. Experimental procedures were in accordance with the Louisiana State University Health Sciences Center guidelines for care and use of laboratory animals (IACUC).

**Fear conditioning**. Fear conditioning training was conducted in a chamber (28 × 28 × 30 cm) with black walls and a 75-mm diameter speaker. The floor was made of stainless-steel rods spaced at 0.5 cm and connected a shock delivery apparatus (Shocker Model H13–15, Coulbourn Instruments, Holliston, MA) (context A). The conditioning apparatus was placed in a sound reducing chamber (typical background noise was 65 dB). The timing and length of both the tones and shocks were adjusted using custom software. All conditioning procedures were conducted during the dark phase of the light/dark cycle, 15 h before slice preparation. Male mice (P18–90) were randomly assigned to one of the following three groups (Fig. 1c). (1) *Fear conditioning*. Animals were positioned in the center of the arena in context A. Following a 2-min acclimation period (baseline activity), mice were exposed to eight pairings of a 10-s tone (3.5 kHz, 75 dB) that co-terminated with a 1-s shock (0.75 mA). The duration between pairings was 30 s. After the last pairing, the animals were left in the conditioning chamber for 2 min. (2) *Unpaired procedure*. Mice were exposed to eight tones alone (30 s interval). Animals were then either returned to their home cage or left in the conditioning chamber for 30 min, and then exposed to a series of eight shocks (every 30 s) in context A. Since these two procedures yielded similar results, the results were pooled. (3) *Naïve animals* were never exposed to the conditioning procedure nor to the conditioning apparatus.

**Fear conditioning and memory retention tests using L7::hM3Dq mice**. Three-month-old male littermates, L7::hM3Dq+ and L7::hM3Dq−, were used for behavioral testing. The experimenter was blind to the genotype of the animals at the time of the test. All animals were identified by marks on the tail and weighed 1 h before the conditioning session. Experiments were conducted on three different litters divided into two sessions. All experiments were video recorded (Windows Media Encoder v9, Microsoft) and stored on a computer for off-line analysis.

*Fear conditioning was* conducted on day 1 (see the fear conditioning procedure). Mice were positioned in the center of conditioning chamber (context A) and conditioned with eight pairings of a tone with a footshock after 2 min acclimation. Animals were returned to their home cage for 30 min and then received an i.p. injection of 0.5 mg/kg CNO (NIMH Chemical Synthesis and Drug Supply Program) or saline as control.

*Cued memory retention* was tested in a chamber with a different context (20 × 35 × 40 cm) having off-white walls and in which the floor was covered in white paper bedding (context B). On day 2, animals were positioned in the center of context B and left to explore the arena for 2 min. They were then presented with eight tones (10 s repeated every 30 s), and left in the arena for 2 min before being returned to their home cage. On day 3, *contextual memory retention* was tested by re-exposing mice to the conditioning chamber (context A) for 2 min without the presentation of tones or shocks.

*Behavioral quantification*. Freezing (immobility) was defined as the absence of movement for at least 1 s and was quantified by the amount of motion that occurred between two successive video frames, using a custom-written program as previously described[70]. The duration of freezing was determined during the 2 min of acclimation and the first 9 s of each tone. Data shown represents the freezing response during the first three tones of the cued memory retention and the entire 2 min of exposure to context A in the contextual memory test.

**Locomotor effects of CNO following L7::hM3DQ activation**. To assess the effectiveness of CNO in activating Gq DREADD receptors in Purkinje cells, mice were injected i.p. with CNO (0.5 mg/kg) 30 min before the open field test. This behavioral assay was conducted in a 35 × 43 × 20 cm glass arena with opaque walls 1 week after fear conditioning testing. Animals were positioned in the center of the arena and left to explore for 10 min. While the entire session was analyzed, only the first 2 min of open-field assay are presented to allow comparison with the 24 h analysis. Analysis of 2 or 10 min sessions showed a similar pattern of ambulatory activity. To evaluate the residual motor effects of CNO 24 h after injection, the distance traveled during the first 2 min in context B prior to the memory retention test was quantified.

All experiments were video recorded and stored on a computer for off-line analysis. To evaluate the ambulatory activity, the travel distance was quantified using open-source tracking software (Kinovea 0.8.24) by an experimenter blind to the genotype of the animals.

**Cerebellar slice preparation and electrophysiology**
*Slice preparation*. Cerebellar slices were prepared as previously described[73,74]. Briefly, P18 to P110 male mice were decapitated and the cerebellum was isolated.

Sagittal slices (300 μm) were cut from the cerebellar vermis using a vibratome (Leica VT1200) in an ice-cold slicing solution (containing in mM: 81.2 NaCl, 2.4 KCl, 23.4 NaHCO$_3$, 1.4 NaH$_2$PO$_4$, 6.7 MgCl$_2$, 0.5 CaCl$_2$, 23.3 glucose, 69.9 sucrose, pH 7.4). Slices were then maintained in aCSF (in mM: 125 NaCl, 2.5 KCl, 26 NaHCO$_3$, 1.25 NaH$_2$PO$_4$, 1 MgCl$_2$, 2 CaCl$_2$, 25 glucose, pH 7.4) saturated with 95% O$_2$, 5% CO$_2$ at room temperature for at least 30 min before recording. All experiments were carried out at near physiological temperature (33–37 °C). Unless otherwise noted, all recordings were obtained in lobules V and VI of the cerebellar vermis with patch pipettes (3–6 MOhm for Purkinje cells, 5–10 MOhm for inter-neurons) pulled from borosilicate capillary glass (Harvard Apparatus, Holliston, MA) with a Narishige PP-830 puller. EPSCs and IPSCs were recorded in the presence of GABA$_A$-R blockers (100 μM PTX + 5 μM SR-95531), and a non-NMDAR inhibitor (5 μM NBQX), respectively. TTX (0.5 μM) was included during recordings of miniature events. Analog signals were filtered at 6 kHz and digitized at 20 kHz (Multiclamp 700A, Axon Instruments). Data were analyzed using Clampex 10.2.0.12 (Axon Instruments).

*Voltage-clamp experiments.* Whole-cell patch clamp recordings were obtained from cerebellar stellate cells held at −60 mV. Stellate cells were identified by their location in the outer two-thirds of the molecular layer and by the presence of spontaneous action potentials in the cell-attached mode. Series resistance, input resistance, and cell capacitance were monitored throughout the experiment and the recordings were discarded if these parameters changed by more than 20%. A pipette solution that contained a low EGTA concentration (in mM: 140 CsCl, 2 NaCl, 0.1 CaCl$_2$, 4 MgATP, 0.5 Cs-EGTA, 1 QX-314, 5 TEA, and 10 HEPES, pH 7.25) was used to evoke eCB release in DSI and DSE experiments. Spontaneous, miniature, and evoked IPSCs were recorded using a high EGTA pipette solution (in mM: 130 CsCl, 2 NaCl, 1 CaCl$_2$, 4 MgATP, 10 Cs-EGTA, 1 QX-314, 5 TEA, and 10 HEPES, pH 7.25).

*Evoked synaptic currents.* EPSCs at the parallel fiber to stellate cell synapse were evoked by stimulating parallel fibers using a monopolar glass electrode (3–6 MOhm) filled with aCSF and recorded in stellate cells. To evoke IPSCs at the stellate-to-stellate cell synapse, a stimulating electrode was placed on the soma of a neighboring presynaptic stellate cell in the upper 2/3 of the molecular layer. Basket cells are located in the inner molecular layer and their axons extend laterally to innervate the soma of Purkinje cells and other neighboring basket cells. It is unlikely that our stimulating electrodes would recruit basket cell axons. For DSE and DSI experiments, the stimulus strength ranged from 2 to 25 V with a duration of 200 μs and was adjusted to evoke near-zero failures. Synaptic currents were evoked at 0.5 Hz and recorded in a stellate cell voltage-clamped at −60 mV for 40 s. The postsynaptic stellate cell was then depolarized to 0 mV for 2 s, and eEPSCs or eIPSCs were recorded at −60 mV for 80 s. For graphical representation, the binning of the time course of DSI was set to 6 s. To determine the amplitude and paired pulse ratio of eIPSCs, inhibitory synaptic currents were evoked by stimulating presynaptic stellate cells with two consecutive stimuli with a 20-ms interval. This paired stimulation was repeated every 3 s, and eIPSCs were recorded in stellate cells using a high EGTA-containing CsCl-based pipette solution. The stimulation intensity was set at 15 ± 2.5 V with a duration of 200 μs and produced a 0.2–0.7 failure rate of the first evoked current. eIPSCs that exhibited multiple peaks in response to one or both of paired stimuli were discarded, and the eIPSC amplitudes from the remaining recordings could be fitted with a single Gaussian, suggesting a single homogenous population. The paired pulse ratio at the stellate-to-stellate cell synapse was calculated as the ratio of the amplitude of the second averaged eIPSC (typically 100 events) divided by the first averaged eIPSC.

*Cell-attached recordings.* Cell-attached recordings were obtained from cerebellar stellate or Purkinje cells (characterized by their large soma and high action potential firing rate) using pipettes filled with aCSF in the presence of extracellular 100 μM PTX and 5 μM SR-95531. In pilot experiments, we observed a change in AP firing with time in a subset of SCs, as shown previously in a study by Alcami and colleagues[75]. Therefore, we chose to record the activity in each stellate cell for 5 min and analyze the action potential frequency during a 1-min recording period. Control stellate cells were recorded prior to AM$^{251}$ application. AM$^{251}$ (5 μM) was then bath applied and extracellular recordings were conducted on several stellate cells 20 min later. We quantified average frequency prior to, and during, AM$^{251}$ application for each animal. A total of 109 stellate cells were recorded and no more than ten cells were included per condition for each animal.

*Voltage-gated calcium currents.* Ca$^{2+}$ currents were evoked by a family of depolarizing steps from the holding potential at −80 mV. The pipette solution contained (in mM): 119 CsCl, 9 EGTA, 10 HEPES, 1.8 MgCl$_2$, 14 Tris-creatine phosphate, 4 ATP-Mg, 0.4 GTP-Na, 10 TEA, 1 QX314, pH 7.3. Ca currents were recorded in ACSF solution that contained 10 mM TEA, 300 nM TTX, 10 μM ZD7288, 1 mM kynurenic acid, 100 μM picrotoxin to block potassium, sodium, and h-currents, and excitatory and inhibitory synaptic currents, respectively. We then applied cadmium (CdCl$_2$, 100 μM), a general calcium channel blocker, and quantified the calcium current as the difference current ($I − I_{Cd}$).

*Optogenetic stimulation.* Optogenetic activation of ChR in Purkinje cells from L7::ChR mice triggered release of endocannabinoids and suppressed spontaneous IPSCs (sIPSCs) in stellate cells, producing heterosynaptic DSI. We chose to record spontaneous IPSCs as this allowed us to sample inhibitory synaptic inputs onto the entire postsynaptic stellate cell. Stellate cells were voltage-clamped at −60 mV and sIPSCs were recorded in the presence of 5 μM NBQX. After a 10-s baseline, the slice was exposed to blue light (10 s, 450–480 nm bandpass filter, Olympus) delivered using a Lambda DG-4 (Sutter Instrument, Novato, CA). Spontaneous IPSCs were recorded for an additional 110 s. Each cell was typically subjected to five trials. We quantified the cumulative amplitude of events in bins of 1.5 s for each sweep and averaged the response from five sweeps for each cell. For representation purposes, binning of the time course was set to 3 s.

Optogenetic conditioning was performed in vitro 5 min after obtaining cerebellar slices from either NOS::ChR or L7::ChR animals. Slices were randomly assigned to a non-photostimulated (control) or photostimulated groups and were maintained in regular aCSF at room temperature. Slices from the photostimulated group were exposed to a 10-s blue light (470 nm, Luxeon K2 LED, Philips Lumileds, San Jose, CA) eight times every 30 s. For experiments in which optogenetic conditioning was conducted in the presence of synaptic blockers, PTX 100 μM + SR-95531 5 μM or NBQX 5 μM + CPP 10 μM were applied 5–10 min before the first light exposure and washed out within 5 min of the last exposure. All slices were then maintained in a light-proof beaker containing O$_2$/CO$_2$-saturated aCSF for at least 2 h.

**Immunohistochemistry.** Cerebellar slices were prepared as described above. Slices were then fixed for 1 h in an ice-cold PBS solution containing 4% paraformaldehyde. After antigen retrieval in a Tris-EDTA solution (pH 9, 4 min at 100 °C) for 5 min, MAGL immunostaining was performed as previously described[76]. The primary antibody used was goat anti-MAGL (1:200) and the secondary antibody was donkey anti-goat dylight 549 (1:100) in Fig. 6. We used two different secondary antibodies, donkey anti-goat dylight 549 or Cy3, for MAGL immunostaining in L7::DREADD-Gq(±) mice. Due to the difference between fluorescent intensity of these two fluorophores, MAGL-ir was normalized to the average MAGL-ir value in L7::DREADD-Gq(−) for each secondary antibody in Fig. 10. For each batch of experiments, primary antibodies were omitted in some incubations to check for non-specific staining.

**Acquisition and analysis of fluorescence images.** Cerebellar sections mounted in Vectashield mounting medium (Vector laboratories) were viewed and imaged using a Leica SP8 a laser scanning confocal microscope (DMI8 CS) with HC PL APO CS2 20x/0.75 DRY objective lenses and controlled by LAS-X software (3.7.2). For each vermal slice, the area imaged included the inner portion of visually identified lobules V/VI and lobules IX/X (see Fig. 6b). Within a set of experiments, all behavioral, staining and imaging procedures were performed in parallel. All images in this study were acquired sequentially with identical settings. For each cerebellar section, a stack of 3–5 focal images near the highest fluorescence level were taken at 5 μm intervals. Mean fluorescence intensity in the granule cell layer or the molecular layer of the cerebellar cortex was quantified using ImageJ software (version 1.53c). The level of fluorescence at the focal plane with the highest mean fluorescence intensity is presented in Figs. 6 and 10, where representative images are displayed with the same contrast and brightness settings. Imaging and analysis were conducted by an experimenter who was blind to the experimental condition.

**Endocannabinoid extraction and measurements.** Vermal slices of the cerebellum were prepared as described above (1200 μm). Lobules I to VI were then dissected, immediately frozen on dry ice and stored for 2 days at −80 °C. The samples were then weighed and crushed in acetonitrile containing a [$^2$H$_8$]2-AG internal standard. The solution was then sonicated at 4 °C for at least 2 h. When cloudy, the solutions were kept at −20 °C for another 2 days. Samples were then centrifuged (3 min at 1500 × g) and the supernatant was transferred to a conical tube. Supernatants were dried using a speedvac for 90 min and separated on a C18 column. 2-AG and [$^2$H$_8$]2-AG were detected at $m/z$ 379 and 387 and quantified by Liquid Chromatography Mass Spectrometry in the Department of Pharmacology, Medical College of WI, Mass Spec core.

**Quantification, statistical analysis, and reproducibility.** Clampfit 9.0 (Axon Instruments) was used for the analysis of spontaneous and miniature IPSCs using a built-in event detection template. The magnitude of DSE and DSI were assessed by measuring the amplitude of the first eEPSC and eIPSC after depolarization relative to the averaged current recorded during a 40-s window preceding the depolarization. The recovery rate of DSE and DSI was obtained by fitting the current amplitude over 80 s following depolarization with a single-exponential function. No statistical method was used to predetermine sample sizes, but they are similar to previous studies[73,74,76,77]. Each data set, including immunohistochemistry and control optogenetic experiments, was obtained from at least three independent replicates from mice originating from at least three different litters and animals were assigned randomly to the different experimental conditions. All values are presented as mean ± SEM, two-sided tests were used, and a P value < 0.05 was

considered as significant. For each statistical analysis, normality and equality of the variances were assessed. All statistical tests were performed on primary data (not normalized), except for the effect on amplitude during DSE and DSI. For detailed statistical analysis, see the Supplementary Table 4. Data are available upon request from the corresponding author.

**Reagents and resources**. A full list of resources tables can be found in the Supplementary information. Further information and requests for resources and reagents should be directed to and will be fulfilled by the lead contact, Dr. S.Q. June Liu (sliu@lsuhsc.edu).

**Reporting summary**. Further information on research design is available in the Nature Research Reporting Summary linked to this article.

## Data availability
Raw and derived data supporting the work presented in this study are available from the corresponding author upon request. Source data are provided with this paper.

## Code availability
The code used for behavioral analysis has already been described in Liu et al.[77], and will be provided by the corresponding author upon reasonable request.

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

## Acknowledgements

This work was supported by National Science Foundation Grant IBN-0344559 and National Institutes of Health Grants NS58867, R01NS106915, and MH095948 (S.Q.J.L.), and an NIH COBRE grant P30 GM106392. We thank Drs. Iaroslav Savtchouk, Charles Nichols, Scott Edwards, Nicholas W. Gilpin, Suzanne Zukin, and Matthew D. Whim for experimental advice and helpful discussions, Dr. Peter J. Winsauer for the use of equipment, and Marilyn Isbell and Dr. Mike Thomas for MS analyses.

## Author contributions

C.J.D. and J.F.-P. performed the experiments and analyzed the data. P.A.K., J.F.-P. and C.J.D. performed the immunohistochemistry experiments. C.J.D. and S.Q.J.L. designed the experiments, interpreted the results, and wrote the manuscript.

## Competing interests

The authors declare no competing interests.
