## [Peer Review File · Nature Communications]

Reviewers' comments:

Reviewer #1 (Remarks to the Author):

This manuscript by Dubois et al. examines the consequences of fear conditioning on eCB control of GABA release. They show that the level of the eCB degrading enzyme MAGL is increased after learning, thus reducing basal eCB level and tonic control of GABA release. They provide evidence that the downregulation of the MAGL is mediated by an increase in GABA release. The experiments are well done and the results are generally convincing. They provide evidence for a learning-dependent modulation of the enzyme involved in the degradation of eCB. While the findings are potentially interesting, some data are not entirely consistent with the proposed model and are hard to put together. I have several points that should be addressed in order to strengthen the conclusions of the study.

The authors present data indicating that fear conditioning increases the level of MAGL in lobule V/VI, thereby decreasing the basal level of the eCB 2-AG and decreasing tonic inhibition of GABA release. The fact that mIPSC frequency is higher after learning and that blockade of CB1R increases mIPSC frequency in naive mice but not after learning is consistent with the idea that there is no tonic control of GABA release by eCB after learning. If this interpretation is correct, one would expect that application of a CB1R agonist should have less effect on GABA release in naive mice (because GABA release is already partially depressed by eCB tone). However, they show that the effect of the CB1R agonist WIN is identical in naive mice and after learning. Similarly, the maximal depression at the peak of DSE is also expected to be reduced if synapses are already depressed by tonic eCB in naive mice. However, they show that the magnitude of the depression is the same in naive and conditioned mice. These data are hard to put together. How do the authors explain that eCB and WIN effect and not smaller in naive mice?

The authors propose that a transient increase in eCB release (through Gq activation) decreases GABA release and reverses the learning-induced change in the cerebellum. From their experiment in fig 7, it is hard to know whether the increase in eCB is indeed the cause of the learning impairment. Do they prevent the learning deficit if a CB1R blocker is injected at the same time as CNO? In addition, their experiment suggests that a transient decrease in GABA release is sufficient to reverse the learning. Can they mimic the effect of CNO by transiently decreasing GABA release (inhibitory DREADD)? They show that 30 min after CNO, IPSC frequency is reduced in a CB1R dependent manner. This manipulation has a lasting effect on learning. Is IPSC frequency still decreased on day 2 or does it come back to control level? Is the learning-induced change in DSE kinetics reversed by this CNO injection?

CNO does not cross the BBB but is converted in vivo into clozapine. This molecule is a known blocker of the leak potassium channels TREK that are expressed in the cerebellum. Is the effect of CNO on IPSC really mediated by DREADD activation? In addition, the experiment with DREADD expression in PC requires additional controls such as the effect of DREADD expression only (without CNO injection) on learning.

How is the fit for the DSE recovery rate estimated? Some fits do not accurately match with the experimental points, especially during the initial recovery phase at the peak of depression (see Fig 1D).

Reviewer #2 (Remarks to the Author):

The endocannabinoid 2-arachidonoylglycerol has been implicated in regulation of certain forms of learning and memory. In this manuscript, the authors report that auditory fear conditioning accelerates degradation of 2-AG through upregulation of the 2-AG hydrolyzing enzyme monoacylglycerol lipase (MAGL). Photo-activation of stellate cells to induced GABA release in cerebellar slices accelerated 2-AG degradation through activation of GABAA receptors. After fear conditioning in vivo, GABA release from stellate cells exhibited long-lasting enhancement, immunoreactivity of MAGL was significantly elevated in cerebellar lobule V/VI and tonic endocannabinoid signaling was accelerated in the same lobules. Enhanced Purkinje cell activity by means of excitatory DREADD after fear conditioning impaired the retention of fear memory. From these results, the authors conclude that enhanced GABA release in the cerebellum after fear conditioning accelerates 2-AG degradation and promotes consolidation of fear memory. The findings are unexpected and may implicate a novel and previously unappreciated role of the endocannabinoid system in memory consolidation. In general, the experiments are performed properly with standard techniques. However, I have several comments and concerns over the results and the interpretation of the data.

Major comments

1. Decay of DSE or DSI was used as a measure of 2-AG-mediated retrograde synaptic modulation. This parameter is affected not only by the degree of 2-AG degradation but also by the intracellular calcium elevation in postsynaptic neuron caused by depolarization. Therefore, it is necessary to check whether fear conditioning and photoactivation of stellate cells affects depolarization-induced calcium transients in postsynaptic neurons. The authors may claim that magnitude of peak DSE/DSI reflects the depolarization-induced calcium elevation in postsynaptic neuron. However, this assumption cannot be applied to cases in which presynaptic CB1 receptors are saturated by 2-AG from postsynaptic neurons following depolarization. The authors should check the calcium concentration of postsynaptic neuron during depolarization directly. Alternatively, they should use shorter and longer depolarizing pulses to induce DSE/DSI and check whether the magnitude of peak DSE/DSI is affected by fear conditioning and photoactivation of stellate cells.
2. If GABA release in the cerebellum is the key factor to induce an increase in 2-AG degradation, bath application of GABA into slices would mimic the results (faster recovery of DSI/DSE and upregulation of MAGL). These experiments would provide further evidence that GABA release induces an increase in 2-AG degradation by upregulation of MAGL.
3. In Figure 3, the effect of the MAGL blocker JZL should be added to confirm that acceleration of DSE decay induced by photostimulation of stellate cells is actually mediated by MAGL.
4. Molecular mechanisms underlying the MAGL upregulation by enhanced GABA-mediated signaling in the cerebellum are not explored in the present study. It is not difficult to check whether inhibitors of protein kinases, phosphatases or protein synthesis block the MAGL upregulation following photoactivation of stellate cells in cerebellar slices. I recommend the author perform such kinds of experiments.

Minor comments

1. In figure 4A,B, how was the stimulus strength of inhibitory inputs determined? Did the authors stimulate single presynaptic axon?

2. The authors described that they stimulated stellate cells (e.g. page 5, line 142). I wonder this stimulation also recruited basket cell axons. In the method, there is no explanation how they stimulated stellate cell axons selectively. It should be mentioned.
3. The expression levels of MAGL in the molecular layer of lobule IX/X is almost the same as in that of lobule VI of paired animals. Does this mean that MAGL is constitutively upregulated in the molecular layer of lobule IX/X?
4. Acquisition of fluorescence images is not mentioned in the method. It is important to show how immunoreactivity of MAGL was measured.
5. Page 3, line 21. "but however" should read as "but".
6. Page 4, line 60. Basket cells also release endocannabinoids (see Beierlein and Regehr, Journal of Neurosci, 2006).
7. Page 4, line 70 and 90. DSE and DSI should be spelled out.
8. Page 5, line 122. The explanation of NOS promotor seems not accurate. Because Purkinje cells also express Chr2 in NOS::Chr2 mice as shown later, this explanation will mislead the readers. Additionally, basket cells also highly express NOS (Kim et al, Cell Rep, 2014).
9. Page 7, line 184. "essay" should be read as "assay".
10. In Figure S4, there are no asterisks for demonstrating PCs.
11. Scale bars of amplitude for evoked responses are missing in many figure panels.

Reviewer #3 (Remarks to the Author):

The present MS investigates the effect of fear conditioning on the endocannabinoid-mediated signaling in the cerebellum. The authors aim at understanding the learning-induced regulation of endocannabinoid production and degradation and how their interference alters behavior. The authors have chosen cued fear conditioning as learning paradigm and, for some reason, they investigated synaptic mechanisms in the cerebellum. Although, the cerebellum has been implicated in a number of higher-order cognitive processes, its primary role is clearly not storing cued fear memory traces. Investigating synaptic alterations in brain areas, which clearly participate in cued memory formation would have been a much better choice. Or, if the authors prefer investigating cerebellar synaptic circuits, a well-known cerebellar related learning would have been preferred. The authors argue that by monitoring the time course of recovery rate of DSE allows them to infer the rate of endocannabinoid degradation. Clearly, no experimental data supports this assumption! There are many other mechanisms that could determine the recovery rate of the DSE! Receptor desensitization/internalization, receptor-effector decoupling, effector inactivation could all contribute to the recovery rate. They also erroneously state that 'recovery was mediated at least in part by degradation of 2-AG because a monoacylglycerol lipase inhibitor, JZL184, prolonged the recovery time'. This is incorrect. The fact that MAGL inhibitor can further prolong the recovery rate does not contain any information regarding the mechanism governing the decay under normal condition. A similar phenomenon/argument is widely accepted in the field of neurotransmission. The fact that the application of glutamate/GABA uptake inhibitors prolong the EPSC/IPSC decay does not tell anything about what shapes the EPSC/IPSC decay. It could be purely a receptor deactivation driven process under normal conditions (which is likely to be the case in most synapses). Indeed, the argument that the recovery rate contains very little information regarding the time course of eCBs in the tissue is

supported by the authors own data. The tau in control is ~ 10 s for DSE, which accelerates to ~ 5 s in paired mice. At the same time, the tau is 30s for DSI in control and accelerates to 15s in paired mice. So what is then the time course of 2-AG in the tissue?

A similar false impression is that the peak of DSI/DSE contains information regarding the 'production of eCB' or 'concentration of eCB'. Why? The authors measure the postsynaptic responses and from their changes they simply infer changes in peak eCB concentration. eCB act on G-protein coupled receptors, which in turn regulate Ca²⁺ channels in a non-linear manner, which regulate release in a highly non-linear fashion. Is not that surprising that the peak DSE and DSI amplitudes are the same when IPSCs are fully mediated by P/Q type Ca²⁺ channels whereas EPSCs are mediated by P/Q- and N-type (and even some R) channels? For these reasons such statements/conclusions cannot be made: 'Therefore, learning did not alter CB1R signaling in stellate cells or endocannabinoid release from Purkinje and stellate cells.' Such conclusion can only be made if we understand the mechanisms governing the peak of DSI/DSE!

The authors declare that stellate cells receive glutamatergic inputs from both climbing and parallel fibers. The climbing fiber input is highly debated. But it is totally irrelevant regarding their experiments, because they did not simulate the climbing fibers to drive stellate cell firing, but instead used the non-physiological ways of activation with Chr2!

Line 129: 'To confirm that GABA release mediates this change' (also written in the abstract)... How could the authors conclude anything about the release of GABA from experiments in which they block postsynaptic GABAA receptors? In the paragraph starting in line 149, the authors again examined 'elevated GABA release' by blocking postsynaptic GABAA receptors! What is the consequence of blocking inhibition in cerebellar slices? The spontaneous firing rate of all neurons should be elevated, resulting in a change in the concentration of all neurotransmitters.

The authors use apparently randomly evoked and spontaneous inhibitory and excitatory postsynaptic currents to monitor eCB effects. They do it even within a single logical set of experiments. What is the rationale of selecting spontaneous or evoked release and recording inhibitory or excitatory PSCs?

Change in the amplitude of evoked IPSC does not mean increased release! Changes in intrinsic (e.g. axonal) excitability cannot be excluded. Such experiments should have been done with paired recordings.

The fitted exponentials in Figure 1E does not seem to be right! The 'paired fitted' cannot be the fit to the data!

Fig 5: Leaving out the primary antibody from the reaction does not hold any information regarding the staining in the presence of the primary antibody. In the latter case, the primary antibody could still bind to unknown targets nonspecifically, resulting in an immunostaining that has nothing to do with the desired protein.

Dear Editor,

Thank you for your suggestions and for sending us the reviewers' comments.

In response we have conducted additional experiments and addressed the reviewers' other concerns by modifying the text of the manuscript. We believe that these changes have significantly strengthened the conclusions and improved the clarity of the manuscript. We are grateful for the reviewers' helpful comments and suggestions.

- 1) We have performed fear conditioning and memory retention experiments on both L7::Gq-DREADD+ and L7::Gq-DREADD- mice without CNO injection, and found that the expression of Gq-DREADD does not impair memory retention (Fig 8E).
- 2) We have tested the effect of a CB1R antagonist on memory impairment following activation of Gq-DREADD in Purkinje cells and found that it prevented the memory deficit (Fig 9).
- 3) We have determined the effect of activation of Gq-DREADD+ in PCs on the learning-induced change in eCB signaling one day after learning. We found that it lowered MAGL expression and reduced spontaneous GABA release due to elevated tonic eCB levels, relative to Gq-DREADD- mice (Fig 10).
- 4) We have quantified voltage-gated Ca currents in molecular layer interneurons from control and fear conditioned mice and detected no difference between them (Fig 3D-F).
- 5) We demonstrate that application of a MAGL inhibitor increased the recovery time of DSE in both photostimulated and non-stimulated MLIs (Fig 4E) using NOS::ChR2 mice.
- 6) We have performed additional recordings, and show the increase in mIPSC frequency following application of a CB1R agonist, WIN, in both naïve and conditioned mice ($mIPSC\ freq_{WIN} - mIPSC\ freq_{no\ WIN}$). These results are presented in Fig 3C and on page 8. Relative change (%), $= (mIPSC\ freq_{WIN} - mIPSC\ freq_{no\ WIN}) / mIPSC\ freq_{no\ WIN}$ is now described on page 5.
- 7) We have modified the Results section to make the statements on page 4 and 6 more precise. We now describe evidence for the non-motor function of the cerebellum in the Discussion on page 11.

We are currently carrying out another study investigating the mechanism underlying the activity-dependent increase in MAGL after learning and photostimulation of molecular layer interneurons. Several experiments suggested by reviewer # 2 have been conducted as part of this new study in which we aim to identify the molecular pathway that upregulates MAGL expression during learning and photo-stimulation of MLIs. Since the present manuscript is already very dense we would prefer not to include these experiments in this submission.

Reviewers' comments:

Reviewer #1 (Remarks to the Author):

This manuscript by Dubois et al. examines the consequences of fear conditioning on eCB control of GABA release. They show that the level of the eCB degrading enzyme MAGL is increased after learning, thus reducing basal eCB level and tonic control of GABA release. They provide evidence that the downregulation of the MAGL is mediated by an increase in GABA release.

The experiments are well done and the results are generally convincing. They provide evidence

for a learning-dependent modulation of the enzyme involved in the degradation of eCB. While the findings are potentially interesting, some data are not entirely consistent with the proposed model and are hard to put together. I have several points that should be addressed in order to strengthen the conclusions of the study.

The authors present data indicating that fear conditioning increases the level of MAGL in lobule V/VI, thereby decreasing the basal level of the eCB 2-AG and decreasing tonic inhibition of GABA release. The fact that mIPSC frequency is higher after learning and that blockade of CB1R increases mIPSC frequency in naive mice but not after learning is consistent with the idea that there is no tonic control of GABA release by eCB after learning.

If this interpretation is correct, one would expect that application of a CB1R agonist should have less effect on GABA release in naive mice (because GABA release is already partially depressed by eCB tone). However, they show that the effect of the CB1R agonist WIN is identical in naive mice and after learning.

Similarly, the maximal depression at the peak of DSE is also expected to be reduced if synapses are already depressed by tonic eCB in naive mice. However, they show that the magnitude of the depression is the same in naive and conditioned mice.

These data are hard to put together. How do the authors explain that eCB and WIN effect and not smaller in naive mice?

The reviewer raises a good point that the CB1R agonist, WIN, is expected to produce a greater decrease in mIPSC frequency after fear conditioning. We have now calculated the difference between mIPSC frequency ($= \text{mIPSC freq}^{\text{WIN}} - \text{mIPSC freq}$) in control and conditioned mice. Indeed, application of WIN reduced mIPSC frequency by 0.25 Hz in control, but 0.66 Hz after fear conditioning ($P = 0.01$; Figure 3C), consistent with the reviewer's prediction. These results are now described in the Results section on page 8.

What we reported in Figure 2E in the earlier version of the manuscript was the proportion of the WIN-induced decrease in mIPSC frequency relative to that before WIN application [$= (\text{mIPSC freq}^{\text{WIN}} - \text{mIPSC freq}) / \text{mIPSC freq}$] in order to account for the difference between basal mIPSC frequency in control and conditioned mice. Although WIN induced a greater decrease in mIPSC frequency ($\text{mIPSC freq}^{\text{WIN}} - \text{mIPSC freq}$) after learning, fear conditioning also elevated basal mIPSC frequency due to lower tonic eCB. Consequently the % change is not different from that in control, suggesting that fear conditioning did not modify CB1 receptor signaling. This result is now described on page 5.

To perform the DSE experiments, we activated multiple parallel fiber inputs. Consequently the EPSC amplitude, and the absolute value of depression (basal EPSC – EPSC at peak depression) varies from cell to cell, and cannot be compared between control and fear conditioned mice. The difference therefore was normalized to the average amplitude of EPSCs before depolarization and is presented as percentage change. This reflects the ability of depolarization-evoked release of eCB to suppress basal EPSC amplitude.

To perform the DSI experiments, we increased the presynaptic stimulation strength such that the probability of triggering eIPSCs was one, and we therefore likely activated multiple inputs. Like DSE, the difference was normalized to the average amplitude of eIPSC before depolarization to quantify the suppression of basal eIPSCs by depolarization-evoked release of eCB in Figure 2A-C. We have shown that putative monosynaptic eIPSC amplitudes evoked with a threshold stimulation strength were significantly increased after fear conditioning (new Figure

5A-B), therefore the net change in amplitude is expected to be greater after fear conditioning. A detailed description of the evoked EPSC and IPSC procedure is now provided in the Methods section on page 23.

The authors propose that a transient increase in eCB release (through Gq activation) decreases GABA release and reverses the learning-induced change in the cerebellum. From their experiment in Fig 7, it is hard to know whether the increase in eCB is indeed the cause of the learning impairment. Do they prevent the learning deficit if a CB1R blocker is injected at the same time as CNO?

We have conducted the experiment suggested by the reviewer and found that administration of a CB1 receptor neutral antagonist immediately after the learning paradigm prevented the impairment of memory consolidation induced by activation of Gq-DREADD. This result reveals that endocannabinoid signaling following activation of Gq-DREADD in Purkinje cells is required for the impairment of memory consolidation *in vivo* and is now shown in Fig 9. We thank the reviewer for this suggestion.

In addition, their experiment suggests that a transient decrease in GABA release is sufficient to reverse the learning. Can they mimic the effect of CNO by transiently decreasing GABA release (inhibitory DREADD)?

This experiment is currently being conducted as part of another ongoing study. Considering the rather large amount of work involved and the scope of this current study, we would prefer to publish this result as a separate body of work.

They show that 30 min after CNO, IPSC frequency is reduced in a CB1R dependent manner. This manipulation has a lasting effect on learning. Is IPSC frequency still decreased on day 2 or does it come back to control level? Is the learning-induced change in DSE kinetics reversed by this CNO injection?

We have shown there is an increase in MAGL-ir (Fig 6), and sIPSC frequency (Fig 5E-F) one day after fear conditioning. To address the reviewer's question, we have quantified spontaneous IPSCs in stellate cells and MAGL-ir in Gq+ and Gq- mice one day after CNO injection. We found that the MAGL-ir level in cerebellar lobule V/VI was lower in Gq+ mice than in Gq- mice (Figure 10A-C). At the cellular level, we show that L7::hM3Dq+ mice exhibited a lower spontaneous IPSC frequency than L7::hM3Dq- mice and application of a CB1R blocker increased the IPSC frequency in L7::hM3Dq+ mice (Figure 10D-E). Therefore, activation of Gq-DREADD in PCs not only impaired memory consolidation, but also reversed the learning-induced increase in endocannabinoid degradation and decrease in endocannabinoid tone in the cerebellum. The results illustrating these long-lasting effects on learning-induced change in eCB signaling are now shown in Figure 10.

CNO does not cross the BBB but is converted *in vivo* into clozapine. This molecule is a known blocker of the leak potassium channels TREK that are expressed in the cerebellum. Is the effect of CNO on IPSC really mediated by DREADD activation?

The reviewer raised an important point about the off-target effects of clozapine or CNO. We applied CNO to slices and tested its effect on IPSC frequency recorded in cerebellar slices. Because a CB1R antagonist blocked the CNO-induced change, we believe that this effect was

indeed mediated by CB1Rs. The point that we want to make is that activation of Gq-DREADD in Purkinje cells can evoke eCB release.

However the *in vivo* situation might be different as indicated by the reviewer. In addition to Gq+ mice, CNO was also injected in Gq- mice to account for any possible off-target effects, such as the blocking TREK channels by clozapine. Because only the Gq+ mice exhibited impaired memory retention it is unlikely that the effect was due to off-target effects of clozapine or CNO. These results are shown in Fig 8E and discussed in the first paragraph on page 9.

In addition, the experiment with DREADD expression in PC requires additional controls such as the effect of DREADD expression only (without CNO injection) on learning.

We have conducted additional experiments to test whether the expression of Gq-DREADD receptors caused an impairment of memory retention. We injected saline into Gq+ and Gq- mice and show that both groups exhibited learning and memory retention that were indistinguishable from wildtype mice, indicating that the activation, rather than expression, of Gq-DREADD is responsible for the deficits in memory consolidation. This result is now shown in Fig 8E.

How is the fit for the DSE recovery rate estimated? Some fits do not accurately match with the experimental points, especially during the initial recovery phase at the peak of depression (see Fig 1D).

The procedure for data fitting is described in the Methods section on page 25. Normalized current amplitude was fitted with a single exponential equation over 30 seconds following the depolarization. In the original manuscript the fitted line in Fig 1D was obtained from the naïve control data, but was mislabeled as unpaired control data. Our intention was to show these two controls have a similar recovery time course. In the revised manuscript, we now present the fitted lines for the paired and unpaired group rather than naïve control. We thank the reviewer for noticing the mistake.

Reviewer #2 (Remarks to the Author):

The endocannabinoid 2-arachidonoylglycerol has been implicated in regulation of certain forms of learning and memory. In this manuscript, the authors report that auditory fear conditioning accelerates degradation of 2-AG through upregulation of the 2-AG hydrolyzing enzyme monoacylglycerol lipase (MAGL). Photo-activation of stellate cells to induced GABA release in cerebellar slices accelerated 2-AG degradation through activation of GABAA receptors. After fear conditioning *in vivo*, GABA release from stellate cells exhibited long-lasting enhancement, immunoreactivity of MAGL was significantly elevated in cerebellar lobule V/VI and tonic endocannabinoid signaling was accelerated in the same lobules. Enhanced Purkinje cell activity by means of excitatory DREADD after fear conditioning impaired the retention of fear memory. From these results, the authors conclude that enhanced GABA release in the cerebellum after fear conditioning accelerates 2-AG degradation and promotes consolidation of fear memory. The findings are unexpected and may implicate a novel and previously unappreciated role of the endocannabinoid system in memory consolidation. In general, the experiments are performed properly with standard techniques. However, I have several comments and concerns over the results and the interpretation of the data.

Major comments

1. Decay of DSE or DSI was used as a measure of 2-AG-mediated retrograde synaptic modulation. This parameter is affected not only by the degree of 2-AG degradation but also by the intracellular calcium elevation in postsynaptic neuron caused by depolarization. Therefore, it is necessary to check whether fear conditioning and photoactivation of stellate cells affects depolarization-induced calcium transients in postsynaptic neurons. The authors may claim that magnitude of peak DSE/DSI reflects the depolarization-induced calcium elevation in postsynaptic neuron. However, this assumption cannot be applied to cases in which presynaptic CB1 receptors are saturated by 2-AG from postsynaptic neurons following depolarization. The authors should check the calcium concentration of postsynaptic neuron during depolarization directly. Alternatively, they should use shorter and longer depolarizing pulses to induce DSE/DSI and check whether the magnitude of peak DSE/DSI is affected by fear conditioning and photoactivation of stellate cells.

The reviewer's point is well taken. Because depolarization activates voltage-gated calcium channels and triggers endocannabinoid release, we quantified the amplitude of depolarization-evoked calcium currents in stellate cells from naïve and fear conditioned animals. We found no difference between groups, suggesting that calcium entry during depolarization is not altered by the conditioning procedure. Thus, it is unlikely that a change in endocannabinoid release in response to membrane depolarization would be responsible for the change in recovery from DSE. This result is now shown in Fig 3D-F.

To show that eCB degradation *via* MAGL is altered following photostimulation, we tested the effect of applying a MAGL inhibitor on the recovery time of DSE. Our results show that the recovery time of DSE was prolonged to 15 sec in MLIs from both control and photostimulated slices. Thus the acceleration of recovery time after photostimulation, like fear conditioning, is at least in part due to an increase in MAGL-dependent 2-AG degradation. This result is now shown in Fig 4.

2. If GABA release in the cerebellum is the key factor to induce an increase in 2-AG degradation, bath application of GABA into slices would mimic the results (faster recovery of DSI/DSE and upregulation of MAGL). These experiments would provide further evidence that GABA release induces an increase in 2-AG degradation by upregulation of MAGL.

We provided two lines of evidence demonstrating the role of GABA signaling in the cerebellum in regulating 2-AG degradation. First, increasing the activity of GABAergic interneurons but not Purkinje cells accelerated the recovery time of DSE, and this effect was prevented by GABA_A receptor blockers. Second, cerebellar slices treated with GABA receptor blockers prolonged the recovery time of DSE in cells from conditioned mice (Figs 4 and 5). Our results suggest that GABA release from interneurons activates GABA_A receptors and this is necessary for enhancing 2-AG degradation in the cerebellum. We think that bath application of GABA onto slices would be less selective and physiological than the approaches we have already used. Unless the reviewer feels strongly about this issue, we would prefer not to revisit this issue with additional experiments.

3. In Figure 3, the effect of the MAGL blocker JZL should be added to confirm that acceleration of DSE decay induced by photostimulation of stellate cells is actually mediated by MAGL.

We have conducted the experiment suggested by the reviewer and found that inhibition of MAGL prolonged the recovery time of DSE to 15 sec in cells in both control and photostimulated slices. Thus acceleration of DSE decay induced by photostimulation of stellate cells is likely mediated by MAGL. This result is shown in Fig 4. We thank the reviewer for the suggestion.

4. Molecular mechanisms underlying the MAGL upregulation by enhanced GABA-mediated signaling in the cerebellum are not explored in the present study. It is not difficult to check whether inhibitors of protein kinases, phosphatases or protein synthesis block the MAGL upregulation following photoactivation of stellate cells in cerebellar slices. I recommend the author perform such kinds of experiments.

We completely agree with the reviewer that it is important to determine the molecular mechanisms underlying the GABA-mediated MAGL upregulation. However one drawback is that inhibitors of protein kinases, phosphatases and protein synthesis affect the function of many proteins, making it difficult to interpret such results.

Minor comments

1. In figure 4A,B, how was the stimulus strength of inhibitory inputs determined? Did the authors stimulate single presynaptic axon?

We placed a stimulating electrode on the somata of a neighboring stellate cell, and extracellularly stimulated this neuron. The strength of the stimulation was adjusted to evoke IPSCs with a 50% success rate. Recordings exhibiting multiple IPSCs in response to each stimulus were discarded. The IPSC amplitudes from the remaining recordings could be fitted with a single Gaussian, suggesting a single homogenous population. However while we aimed to stimulate a single presynaptic neuron, we cannot rule out the possibility that we might activate additional axons. This part of the Methods section has been updated (page 23).

2. The authors described that they stimulated stellate cells (e.g. page 5, line 142). I wonder this stimulation also recruited basket cell axons. In the method, there is no explanation how they stimulated stellate cell axons selectively. It should be mentioned.

The stimulating electrode was placed on the somata of a neighboring stellate cell in the upper 2/3 of the molecular layer. Basket cells are located in the inner molecular layer and their axons extend laterally to innervate the soma of Purkinje cells and other neighboring basket cells. It is therefore unlikely that our stimulating electrodes would recruit basket cell axons. This is now described in the methods section on page 23.

3. The expression levels of MAGL in the molecular layer of lobule IX/X is almost the same as in that of lobule VI of paired animals. Does this mean that MAGL is constitutively upregulated in the molecular layer of lobule IX/X?

The expression level of MAGL in the molecular layer of lobule IX/X is elevated relative to that of lobule V/VI (Figure 6) in naïve animals. This suggests that there are lower levels of tonic endocannabinoids in lobule IX/X. Indeed we found a higher mIPSC frequency (Figure S5c), and a lack of effect of the CB1 receptor antagonist on mIPSC frequency in lobules IX/X (Figure 7D). Therefore these results suggest that MAGL is constitutively upregulated in the molecular layer

of lobule IX/X, but the underlying mechanism remains to be determined. This may explain the discrepancy in tonic endocannabinoid levels in the cerebellar cortex with application of CB1 receptor antagonists, which increases spontaneous GABA release in some studies¹, but not others². This point is now discussed on page 12.

4. Acquisition of fluorescence images is not mentioned in the method. It is important to show how immunoreactivity of MAGL was measured. This point is now described in the Methods section on page 25.

5. Page 3, line 21. “but however” should read as “but”. This is now corrected.

6. Page 4, line 60. Basket cells also release endocannabinoids (see Beierlein and Regehr, Journal of Neurosci, 2006). This is now corrected.

7. Page 4, line 70 and 90. DSE and DSI should be spelled out. We have made the changes as suggested.

8. Page 5, line 122. The explanation of NOS promotor seems not accurate. Because Purkinje cells also express ChR2 in NOS::ChR2 mice as shown later, this explanation will mislead the readers. Additionally, basket cells also highly express NOS (Kim et al, Cell Rep, 2014).

The inaccuracy of the statement has been clarified.

9. Page 7, line 184. “essay” should be read as “assay”. Corrected.

10. In Figure S4, there are no asterisks for demonstrating PCs. This is now corrected.

11. Scale bars of amplitude for evoked responses are missing in many figure panels.

All evoked currents were normalized and we therefore show the scale bar for the time axis only.

Reviewer #3 (Remarks to the Author):

The present MS investigates the effect of fear conditioning on the endocannabinoid-mediated signaling in the cerebellum. The authors aim at understanding the learning-induced regulation of endocannabinoid production and degradation and how their interference alters behavior. The authors have chosen cued fear conditioning as learning paradigm and, for some reason, they investigated synaptic mechanisms in the cerebellum. Although, the cerebellum has been implicated in a number of higher-order cognitive processes, its primary role is clearly not storing cued fear memory traces. Investigating synaptic alterations in brain areas, which clearly participate in cued memory formation would have been a much better choice. Or, if the authors prefer investigating cerebellar synaptic circuits, a well-known cerebellar related learning would have been preferred.

The reviewer questions our choice of the cerebellum as a model structure for cued fear conditioning. Clinical studies show that the cerebellum is critically involved in cognitive and emotional regulation. There is strong evidence supporting a non-motor role for the cerebellum in the consolidation of fear memory and social behavior and autism³⁻⁵. This could result from

information processing within the cerebellum and/or the extensive connections of the cerebellum with cortical and sub-cortical regions. Cerebellar non-motor function is clinically important, and understanding the mechanism underlying emotional learning and memory in the cerebellum could suggest novel treatments for psychiatric disorders. This is an upcoming but currently understudied research area.

Cerebellum and fear conditioning. It is now recognized that several brain regions are involved in the formation of associative fear memories. Growing evidence shows that the cerebellum is required for the consolidation of cued fear memory in rats³, zebrafish⁶, goldfish⁷ and humans^{8,9}, and undergoes learning-induced synaptic and structural plasticity¹⁰⁻¹². Despite the mounting evidence for its role in emotional memory, the cerebellum is often overlooked as an important site of plasticity related to consolidation of conditioned fear memory¹³. Our results show that selective activation of Gq-DREADD in cerebellar Purkinje cells attenuated memory consolidation thus clearly demonstrating the important role of the cerebellum in emotional memory formation. Our findings present a novel mechanism for the regulation of endocannabinoid signaling that is critical for memory consolidation and this is likely to have broad implications for other brain regions. This highlights the **novelty** of our study and the **need for more studies investigating the non-motor function of the cerebellum**, including cued fear learning. We have included additional discussion on the non-motor function of the cerebellum page 11 and 12.

The authors argue that by monitoring the time course of recovery rate of DSE allows them to infer the rate of endocannabinoid degradation. Clearly, no experimental data supports this assumption! There are many other mechanisms that could determine the recovery rate of the DSE! Receptor desensitization/internalization, receptor-effector decoupling, effector inactivation could all contribute to the recovery rate. They also erroneously state that 'recovery was mediated at least in part by degradation of 2-AG because a monoacylglycerol lipase inhibitor, JZL184, prolonged the recovery time'. This is incorrect. The fact that MAGL inhibitor can further prolong the recovery rate does not contain any information regarding the mechanism governing the decay under normal condition. A similar phenomenon/argument is widely accepted in the field of neurotransmission. The fact that the application of glutamate/GABA uptake inhibitors prolong the EPSC/IPSC decay does not tell anything about what shapes the EPSC/IPSC decay. It could be purely a receptor deactivation driven process under normal conditions (which is likely to be the case in most synapses). Indeed, the argument that the recovery rate contains very little information regarding the time course of eCBs in the tissue is supported by the authors own data. The tau in control is ~10s for DSE, which accelerates to ~5s in paired mice. At the same time, the tau is 30s for DSI in control and accelerates to 15s in paired mice. So what is then the time course of 2-AG in the tissue?

Evidence for activity-dependent enhancement of 2-AG degradation. The reviewer made the point that a change in recovery time following learning is not sufficient to conclude that 2-AG degradation is affected. In our study, we provided three lines of evidence for an activity-dependent increase in 2-AG degradation: **1)** an increase in recovery rate; **2)** an increase in MAGL expression; **3)** a decrease in tonic 2-AG levels using mass spectrometry and electrophysiological techniques. Based on these data we conclude that 2-AG degradation is enhanced.

Role of endocannabinoid degradation in determining recovery rate of DSE. Both inhibition and deletion of MAGL prolongs the recovery time of DSE and DSI in the cerebellum and other

brain regions^{14,15}, suggesting degradation of 2-AG by MAGL is the primary known mechanism that governs the recovery rate from depression during DSE and DSI in the cerebellum. In theory, *receptor desensitization/ internalization, receptor-effector decoupling, effector inactivation could all contribute to the recovery rate*, as suggested by the reviewer. However the acceleration of recovery time we observed in paired animals and following photostimulation is unlikely due to a change in eCB receptors and its effector, as inhibition of MAGL prolonged the time course and abolished the difference between paired and controls (Fig 1E-F), and we did not detect any changes in CB1R signaling in paired animals (Fig 3A-C). Together with the increase in MAGL expression these results suggest an increase in 2-AG degradation following fear conditioning.

The reviewer also questions **why recovery time of DSE is different from that of DSI**. MAGL is expressed at high levels in granule cells and a lower level in glial cells, but absent in molecular layer interneurons¹⁴. The heterogeneous expression pattern of MAGL in the cerebellar cortex is likely to contribute to the distinct time course of 2-AG at different synapses. At excitatory synapses, 2-AG release from postsynaptic stellate cells is likely to be rapidly removed and degraded by MAGL in presynaptic parallel fiber terminals and processes of Bergmann glial cells during DSE¹⁴. In contrast stellate cells do not express MAGL and lack presynaptic MAGL at inhibitory synapses¹⁴, and therefore it takes longer for 2-AG to be removed by the nearby processes of Bergmann glial cells and parallel fibers during DSI. These points are now discussed in the Results section on page 4 and 5.

A similar false impression is that the peak of DSI/DSE contains information regarding the 'production of eCB' or 'concentration of eCB'. Why? The authors measure the postsynaptic responses and from their changes they simply infer changes in peak eCB concentration. eCB act on G-protein coupled receptors, which in turn regulate Ca²⁺ channels in a non-linear manner, which regulate release in a highly non-linear fashion. Is not that surprising that the peak DSE and DSI amplitudes are the same when IPSCs are fully mediated by P/Q type Ca²⁺ channels whereas EPSCs are mediated by P/Q- and N-type (and even some R) channels? For these reasons such statements/conclusions cannot be made: 'Therefore, learning did not alter CB1R signaling in stellate cells or endocannabinoid release from Purkinje and stellate cells.' Such conclusion can only be made if we understand the mechanisms governing the peak of DSI/DSE!

Because the peak of DSE is reduced by DAGL inhibitors and CB1R blockers (references^{1,16-19} and Figures 1B and 1G), a change in 2-AG production or CB1R signaling would be expected to alter the amplitude of DSE. In our study, we compared the peak of DSE (or DSI) in conditioned mice with controls, and did not detect any change (Figures 1 and 2). We determined the effects of a CB1R agonist on spontaneous GABA release, and found no difference between control and conditioned mice, suggesting that CB1R signaling was not altered. Because endocannabinoid production during DSE/DSI depends on calcium entry in response to depolarization, we now quantified calcium currents in stellate cells and show that fear conditioning did not alter calcium currents (Fig 3D-F). We have now rephrased the sentence to "Therefore learning did not modify tonic CB1R signaling or depolarization-evoked Ca²⁺ currents that trigger endocannabinoid release from stellate cells." (Results section, page 5 in our revised manuscript).

The question raised by the reviewer, why peak suppression during DSE is similar to DSI given that different subtypes of Ca channels are present at excitatory and inhibitory terminals, is interesting, but beyond the scope of this study.

The authors declare that stellate cells receive glutamatergic inputs from both climbing and parallel fibers. The climbing fiber input is highly debated. But it is totally irrelevant regarding their experiments, because they did not simulate the climbing fibers to drive stellate cell firing, but instead used the non-physiological ways of activation with ChR2!

It has been shown that glutamate released from climbing fibers can activate AMPA and NMDA receptors on stellate cells through spillover-mediated transmission in vivo as well as in vitro²⁰⁻²². Associative learning is known to activate both parallel and climbing fibers²³, producing strong depolarization in molecular layer interneurons. Our goal was to test whether stellate cell activity was sufficient to induce the change. Since co-stimulation of parallel and climbing fibers would activate many types of cells, we therefore chose to activate molecular layer interneurons with ChR2. We now include references for climbing fiber inputs and soften the statement to “Cerebellar stellate cells can be depolarized by stimulation of parallel and climbing fibers”, as this information provides a context for our photostimulation experiment. If the reviewer feels strongly about this point we will remove the sentence.

Line 129: ‘To confirm that GABA release mediates this change’ (also written in the abstract)... How could the authors conclude anything about the release of GABA from experiments in which they block postsynaptic GABA_A receptors? In the paragraph starting in line 149, the authors again examined ‘elevated GABA release’ by blocking postsynaptic GABA_A receptors! What is the consequence of blocking inhibition in cerebellar slices? The spontaneous firing rate of all neurons should be elevated, resulting in a change in the concentration of all neurotransmitters.

We reached the conclusion based on several lines of evidence, rather than the effect of GABA_A receptor inhibitor alone.

1. Stellate cells are GABAergic interneurons and we have shown that photostimulation of these neurons causes a change in the DSE recovery rate.
2. Inhibition of GABA_A receptors during photostimulation blocked the acceleration in the recovery rate of DSE, further supporting the idea that activation of GABA_A receptors is required to induce the change in 2-AG degradation.
3. As suggested by the reviewer, blockade of GABA receptors could increase Purkinje cell activity. Our control experiment using Purkinje cell-specific ChR2 shows that photostimulation of Purkinje cells is not sufficient to accelerate DSE recovery time.
4. Glutamate receptor blockers did not have any effect, indicating it is unlikely that an increase in glutamate receptor activation is responsible for the change during photo-stimulation.

The reviewer’s point that blocking GABA_A receptors can increase the spontaneous activity of other neurons is well taken. Although the mechanism underlying the regulation of DSE recovery time following blocking GABA receptors in conditioned mice might be complicated, this result supports the idea that inhibitory interneuron activity can regulate eCB degradation. We have taken the reviewers comments into consideration and rephrased our question (“To test whether GABA_A receptor activity enhanced 2-AG degradation after learning”) and the conclusion regarding the prolonged DSE recovery time after incubation of cerebellar slices from conditioned mice with GABA_A receptor blockers (“Thus inhibition of GABA_A receptor leads to

suppression of 2-AG degradation. It is also possible that spontaneous activity of other neurons due to blocking GABA_A receptors contributes to the change”). Given that both photostimulation of MLIs and fear conditioning increased GABA release and accelerated DSE recovery rate, our results suggest that a high level of MLI activity promotes 2-AG degradation. These changes are made in the Results section on page 6.

The authors use apparently randomly evoked and spontaneous inhibitory and excitatory postsynaptic currents to monitor eCB effects. They do it even within a single logical set of experiments. What is the rationale of selecting spontaneous or evoked release and recording inhibitory or excitatory PSCs?

DSE vs DSI. Because deletion of MAGL leads to a prolonged recovery time of both DSE and DSI in synapse-independent manner in Purkinje cells¹⁴, we tested the effect of learning on both DSE and DSI. Furthermore, we induced DSI using two methods, depolarization of stellate cells and photoactivation of Purkinje cells. Again, a change in 2-AG degradation is expected to alter the time course of both forms of DSI regardless of the cell type that releases 2-AG. The rationale is now presented in the Results section on pages 4 and 5.

Evoked vs spontaneous synaptic currents. For DSE and DSI in response to somatic depolarization in stellate cells we measured evoked synaptic currents to detect synapse specific suppression as described previously^{19,24}. However, to test the effect of eCB released from Purkinje cells on inhibitory transmission onto stellate cells, we used L7::ChR2 mice in which ChR2 is expressed in Purkinje cells. Photostimulation evoked release eCB from soma and dendrites of all PCs, which is likely to affect many presynaptic terminals. We therefore chose to record spontaneous IPSCs as this allows us to sample inhibitory synaptic inputs onto the entire postsynaptic stellate cell. This is now described in the methods section on page 24. To determine the effects of CB1R agonist and antagonists on GABA release, we measured miniature IPSCs as these experiments required stable long-term recordings (1-2 hours) that are more difficult to obtain using evoked IPSCs.

Change in the amplitude of evoked IPSC does not mean increased release! Changes in intrinsic (e.g. axonal) excitability cannot be excluded. Such experiments should have been done with paired recordings.

We tested GABA release using two experimental approaches. *First*, we show that learning **increases mIPSC frequency**. *Second*, using evoked IPSCs in which we placed a stimulating electrode on a neighboring stellate cell to evoke GABA release. However we cannot rule out the possibility that we also stimulated an axon in proximity to the stimulated neuron. We quantified **the IPSC amplitude and PPR** and found **an increase in IPSC amplitude and decrease in PPR**. Together these results suggest that learning increases GABA release, although we cannot rule out the contribution of changes in intrinsic excitability to the increase in evoked EPSC amplitude. Our results are consistent with a previous study demonstrating that fear conditioning increases mIPSC frequency recorded in Purkinje cells¹¹. We now presented learning-induced changes in mIPSC and sIPSC frequency in Figure 5 C-F rather than in supplemental figures.

The fitted exponentials in Figure 1E does not seem to be right! The ‘paired fitted’ cannot be the fit to the data!

The fitted lines shown Fig 1E was for control and paired without JZL to illustrate changes

induced by the addition of JZL. We apologize for the confusion and have now removed these fitted lines.

Fig 5: Leaving out the primary antibody from the reaction does not hold any information regarding the staining in the presence of the primary antibody. In the latter case, the primary antibody could still bind to unknown targets nonspecifically, resulting in an immunostaining that has nothing to do with the desired protein.

We agree with the reviewer that a lack of staining using secondary antibody alone does not indicate whether the primary antibody is specific for MAGL. However this is a control for non-specific binding of secondary antibody. We have now removed “No primary” controls from Figure 6 and described the no primary control experiments in the methods section on pages 24-25.

References

1. Kreitzer, A. C., Carter, A. G. & Regehr, W. G. Inhibition of interneuron firing extends the spread of endocannabinoid signaling in the cerebellum. *Neuron* **34**, 787–796 (2002).
2. Galante, M. & Diana, M. A. Group I Metabotropic Glutamate Receptors Inhibit GABA Release at Interneuron-Purkinje Cell Synapses through Endocannabinoid Production. *J. Neurosci.* **24**, 4865–4874 (2004).
3. Sacchetti, B., Baldi, E., Lorenzini, C. A. & Bucherelli, C. Cerebellar role in fear-conditioning consolidation. *Proc. Natl. Acad. Sci. U. S. A.* **99**, 8406–8411 (2002).
4. Carta, I., Chen, C. H., Schott, A. L., Dorizan, S. & Khodakhah, K. Cerebellar modulation of the reward circuitry and social behavior. *Science* **363**, (2019).
5. Stoodley, C. J. *et al.* Altered cerebellar connectivity in autism and cerebellar-mediated rescue of autism-related behaviors in mice. *Nat. Neurosci.* **20**, 1744–1751 (2017).
6. Matsuda, K., Yoshida, M., Kawakami, K., Hibi, M. & Shimizu, T. Granule cells control recovery from classical conditioned fear responses in the zebrafish cerebellum. *Sci. Rep.* **7**, 11865 (2017).
7. Yoshida, M., Okamura, I. & Uematsu, K. Involvement of the cerebellum in classical fear conditioning in goldfish. *Behav. Brain Res.* **153**, 143–148 (2004).
8. Lange, I. *et al.* The anatomy of fear learning in the cerebellum: A systematic meta-analysis. *Neurosci. Biobehav. Rev.* **59**, 83–91 (2015).
9. Utz, A. *et al.* Cerebellar vermis contributes to the extinction of conditioned fear. *Neurosci. Lett.* **604**, 173–177 (2015).
10. Zhu, L., Scelfo, B., Tempia, F., Sacchetti, B. & Strata, P. Membrane excitability and fear conditioning in cerebellar Purkinje cell. *Neuroscience* **140**, 801–810 (2006).
11. Scelfo, B., Sacchetti, B. & Strata, P. Learning-related long-term potentiation of inhibitory synapses in the cerebellar cortex. *Proc. Natl. Acad. Sci. U. S. A.* **105**, 769–774 (2008).
12. Ruediger, S. *et al.* Learning-related feedforward inhibitory connectivity growth required for memory precision. *Nature* **473**, 514–518 (2011).
13. Apps, R. & Strata, P. Neuronal circuits for fear and anxiety - the missing link. *Nat. Rev. Neurosci.* **16**, 642 (2015).
14. Tanimura, A. *et al.* Synapse type-independent degradation of the endocannabinoid 2-arachidonoylglycerol after retrograde synaptic suppression. *Proc. Natl. Acad. Sci. U. S. A.* **109**, 12195–12200 (2012).
15. Pan, B. *et al.* Blockade of 2-arachidonoylglycerol hydrolysis by selective monoacylglycerol lipase inhibitor 4-nitrophenyl 4-(dibenzo[d][1,3]dioxol-5-yl(hydroxy)methyl)piperidine-1-carboxylate (JZL184) Enhances retrograde endocannabinoid signaling. *J. Pharmacol. Exp. Ther.* **331**, 591–597 (2009).
16. Hashimoto-dani, Y. *et al.* Acute inhibition of diacylglycerol lipase blocks endocannabinoid-mediated retrograde signalling: evidence for on-demand biosynthesis of 2-arachidonoylglycerol. *J. Physiol.* **591**, 4765–4776 (2013).
17. Jain, T., Wager-Miller, J., Mackie, K. & Straiker, A. Diacylglycerol lipase α (DAGL α) and DAGL β cooperatively regulate the production of 2-arachidonoyl glycerol in autaptic hippocampal neurons. *Mol. Pharmacol.* **84**, 296–302 (2013).
18. Yoshino, H. *et al.* Postsynaptic diacylglycerol lipase mediates retrograde endocannabinoid suppression of inhibition in mouse prefrontal cortex. *J. Physiol.* **589**, 4857–4884 (2011).
19. Beierlein, M. & Regehr, W. G. Local interneurons regulate synaptic strength by retrograde release of endocannabinoids. *J. Neurosci. Off. J. Soc. Neurosci.* **26**, 9935–9943 (2006).
20. Satake, S. *et al.* Characterization of AMPA receptors targeted by the climbing fiber transmitter mediating presynaptic inhibition of GABAergic transmission at cerebellar interneuron-Purkinje cell synapses. *J. Neurosci. Off. J. Soc. Neurosci.* **26**, 2278–2289 (2006).
21. Szapiro, G. & Barbour, B. Multiple climbing fibers signal to molecular layer interneurons exclusively via glutamate spillover. *Nat. Neurosci.* **10**, 735–742 (2007).
22. Jirenhed, D.-A., Bengtsson, F. & Jörntell, H. Parallel fiber and climbing fiber responses in rat cerebellar cortical neurons in vivo. *Front. Syst. Neurosci.* **7**, 16 (2013).
23. Freeman, J. H. The ontogeny of associative cerebellar learning. *Int. Rev. Neurobiol.* **117**, 53–72 (2014).
24. Diana, M. A. & Marty, A. Endocannabinoid-mediated short-term synaptic plasticity: depolarization-induced suppression of inhibition (DSI) and depolarization-induced suppression of excitation (DSE). *Br. J. Pharmacol.* **142**, 9–19 (2004).

REVIEWER COMMENTS

Reviewer #1 (Remarks to the Author):

The authors have performed several additional experiments in order to address my and other referees' concerns. These additions clearly strengthen the main conclusion and the manuscript has been nicely improved. I have no further comment.

Reviewer #2 (Remarks to the Author):

The authors have addressed some of my comments satisfactorily. I agree with the authors that the elucidation of molecular mechanisms underlying the phenomenon will produce a large amount of data and may be suitable for a separate paper (my major comment 4). I have minor comments for further improving the manuscript.

(1) The current manuscript is partly prepared in the style of Neuron/Cell but not in the style of Nature Communications.

(2) Page 4, line 65-67 "...as deletion or inhibition of DAGL (diacylglycerol lipase, a rate-limiting enzyme in the production of 2-AG) and CB1R reduces the amplitude of the suppression,"

The following paper should be cited here.

"Tanimura A et al.(2010) The endocannabinoid 2-arachidonoylglycerol produced by diacylglycerol lipase α mediates retrograde suppression of synaptic transmission. Neuron 65: 320-327"

Reviewer #3 (Remarks to the Author):

The authors argue that the cerebellum has many non-motor functions. The reviewer has never debated this. The reviewer instead pointed out that there are brain regions that are well known to have a critical role in cued fear conditioning and there are many well-known cerebellar functions. It would have been more convincing to start with a function (e.g. motor learning) that is widely accepted as a cerebellar function and only after that look at a behavior cerebellar components is more debated. The authors do not provide any evidence supporting their argument that the time course of DSI/E is mediated by the degrading enzyme. The reviewer emphasizes again that the fact that the inhibition of the degrading enzyme prolongs the decay of DSI/E contains no information on what mediated the decay before the drug was applied.

The authors argue that understanding why the suppression of DSI and DSE is similar although they are mediated by different Ca channels is out of the scope of the present study. The reviewer would agree, but the fact that it is highly unexpected, the authors should have performed more experiments to ensure that their conclusion is the most likely one.

GABA release: here the reviewer pointed out the importance of wording. If there is no evidence for change in release, please do not use it.

The reviewer raised a point that the authors apparently randomly used evoked and spontaneous EPSC/IPCSs in the study. In their rebuttal, they provided explanation of why they believe that using once this and then the other is correct. The reviewer feels that they should have performed experiments consistently using the same type of postsynaptic response.

We were happy to learn that two reviewers are satisfied with our revision, and have now made changes to the text that address the new comments noted in the second round of review. Changes are highlighted in blue in the manuscript.

Reviewer #2 (Remarks to the Author):

The authors have addressed some of my comments satisfactorily. I agree with the authors that the elucidation of molecular mechanisms underlying the phenomenon will produce a large amount of data and may be suitable for a separate paper (my major comment 4).

I have minor comments for further improving the manuscript.

(1) The current manuscript is partly prepared in the style of Neuron/Cell but not in the style of Nature Communications.

The manuscript is now written in the style of Nature communications.

(2) Page 4, line 65-67 "...as deletion or inhibition of DAGL (diacylglycerol lipase, a rate-limiting enzyme in the production of 2-AG) and CB1R reduces the amplitude of the suppression,"

The following paper should be cited here.

"Tanimura A et al.(2010) The endocannabinoid 2-arachidonoylglycerol produced by diacylglycerol lipase α mediates retrograde suppression of synaptic transmission. Neuron 65: 320-327"

We have included Tanimura A et al. (2010) as the reference for this statement.

Reviewer #3 (Remarks to the Author):

The authors argue that the cerebellum has many non-motor functions. The reviewer has never debated this. The reviewer instead pointed out that there are brain regions that are well known to have a critical role in cued fear conditioning and there are many well-known cerebellar functions. It would have been more convincing to start with a function (e.g. motor learning) that is widely accepted as a cerebellar function and only after that look at a behavior cerebellar components is more debated.

While several brain regions are well known to have a critical role in cued fear conditioning, a recent review article highlighted the need to move beyond a traditional amygdala- and prefrontal cortex-focused investigation of fear memory regulation in an effort to identify novel neurocircuitry and pharmacological targets for therapeutic interventions for anxiety and trauma-related disorders (Lebois et al, 2019). Our results which show that reducing endocannabinoid signaling in the cerebellum is required for the consolidation of fear memory identify the cerebellum and eCB degradation as pharmacological targets for therapeutic interventions. Our finding that GABAergic interneurons can regulate endocannabinoid signaling reveals a critical link between these two major neural transmitter systems and expands our understanding of the brain regions involved in cued fear conditioning.

We disagree with the reviewer's comment that "It would have been more convincing to start with a function (e.g. motor learning) that is widely accepted as a cerebellar function and only after that look at a behavior cerebellar components is more debated". There is no a priori reason to think that cerebellar motor learning would use the same mechanism as fear conditioning. In fact we now know that the neural basis for the diverse functions of the cerebellum partially lies in its structural compartmentalization in a lobule specific manner (e.g. vermal lobule V/VI for emotional memory formation; lobule XI/X for motor learning; Crus II for social interactions and vermal lobule 7 for repetitive behavior). At the cellular level, we have shown that tonic endocannabinoid signaling was detected in lobule V/VI, but not in lobule IX/X. Therefore experience-dependent plasticity is expected to be lobule specific and the functional consequences of neural plasticity are likely to be behavioral phenotype specific. Cerebellar motor has certainly been well studied but there is a strong interest in the non-motor function of the cerebellum and the clinical implications (for example as evident from the large number of well attended symposiums on this topic during the 2019 SfN meeting).

We have made the following change:

Line 414: We now include these sentences: "Considering the neural basis for the diverse functions of the cerebellum partially lies in its structural compartmentalization (e.g. vermal lobule V/VI for emotional memory formation; lobule IX/X for motor learning), experience-dependent plasticity is also expected to be lobule specific. Indeed, we and others show that associative fear learning induces LTP at both excitatory and inhibitory synapses and promotes MAGL expression in lobule V/VI, not in lobule IX/X. In contrast motor learning selectively enhances feed-forward inhibitory connectivity in lobule IX/X while fear conditioning increases inhibition growth in lobule V/VI (Ruediger et al, 2011)."

The authors do not provide any evidence supporting their argument that the time course of DSI/E is mediated by the degrading enzyme. The reviewer emphasizes again that the fact that the inhibition of the degrading enzyme prolongs the decay of DSI/E contains no information on what mediated the decay before the drug was applied.

There appears to be some miscommunication on our part with regard to this subject. We do not claim that the time course of DSI/E is solely mediated by the degrading enzyme. However there is overwhelming evidence from the literature, in addition to the contributions from this study, that support the idea that degradation enzymes accelerate the time course of DSI/E in cerebellar neurons, while other factors can influence the recovery time. The question we asked in this study is how learning regulates 2-AG degradation.

Evidence supporting the idea that the degrading enzyme accelerates the time course of DSI/E include **1)** deletion of MAGL in cerebellar neurons prolongs the recovery rate of DSE and DSI in cerebellar neurons (Tanimura et al, 2012); **2)** learning elevates MAGL expression and accelerates the recovery time of DSE and DSI. We and others show that inhibition of MAGL increases the duration of DSE/I (Straiker et al., 2011; Zhong et al., 2011; Liu et al; 2016; Tanimura et al., 2012; Chen et al., 2016). These results indicate that 2-AG degradation can accelerate the time course of DSI/E, although the mechanism by which 2-AG degradation shapes the DSE/I time course might be more complicated than just prolonging the activation of CB1Rs, as suggested by the reviewer. The duration of DSI can also be influenced by

presynaptic mGluRs in neurons in the hypothalamus, when both excitatory and inhibitory inputs are activated (Colmers and Bains, 2018).

Evidence for learning-induced increase in 2-AG degradation. This study reports a change in 2-AG degradation following fear conditioning. Acceleration of the time course of DSE/I, is one of several parameters that we determined. We show an increased expression level of the 2-AG degrading enzyme MAGL and reduced tonic 2-AG levels. One experiment that the reviewer has an issue with is the effects of a MAGL inhibitor on the time course of DSE. The logic behind the experiment is the following: if learning accelerates DSE recovery via a MAGL-independent mechanism, the recovery time should remain faster in conditioned mice than controls in the presence of a MAGL inhibitor. In contrast we found that inhibition of MAGL abolished the difference in the recovery rate of DSE, consistent with learning-enhanced 2-AG degradation rather than an MAGL-independent mechanism. This is a well accepted approach and reviewer #2 requested we do this experiment to show that increased degradation is responsible for accelerated recovery time 3 hours after photostimulation of molecular layer interneurons (first round of reviews). Using multiple approaches, we believe we have convincingly demonstrated that learning enhances 2-AG degradation.

Therefore, we respectfully disagree with reviewer #3 about the lack of evidence supporting the argument that the time course of DSI/E is mediated by the degrading enzyme. However we recognize that some of these points were not clearly communicated and we have made following changes on page 4 (line 63, and line 84-93) to explain this more accurately and explicitly in the revised MS.

Line 63: The sentence, “2-AG is removed via degradation by monoacylglycerol lipase, which leads to the recovery of synaptic transmission” has been changed to “2-AG is removed via degradation by monoacylglycerol lipase, which accelerates the recovery of synaptic transmission”.

Line 84: We now include these sentences: “While evidence supports that degradation enzymes accelerate the time course of DSE in cerebellar neurons, other factors can also influence the recovery time as DSE still slowly recovered when MAGL was deleted or inhibited^{4,5}. We reasoned that if learning accelerates DSE recovery *via* a MAGL-independent mechanism, the recovery time should remain faster in conditioned than control mice in the presence of a MAGL inhibitor. In contrast, we found that inhibition of monoacylglycerol lipase (MAGL) abolished the difference in the recovery rate of DSE.....”.

The authors argue that understanding why the suppression of DSI and DSE is similar although they are mediated by different Ca channels is out of the scope of the present study. The reviewer would agree, but the fact that it is highly unexpected, the authors should have performed more experiments to ensure that their conclusion is the most likely one.

There is an extensive literature on DSE and DSI in the cerebellum. We compared the magnitude of DSE with DSI we found with that reported in 16 publications (see Figure below). Five studies quantify both DSE and DSI (Tanimura et al 2011 and 2012; Liu et al 2016; Zsabo et al 2006; and Hashimoto et al 2013) and find the magnitude of DSE/DSI in Purkinje cells to be 50/50, 50/50, 60/40, 70/40, and 50/40 (%/%), respectively. The average magnitude of suppression at PF-PC synapses (DSE) was 57% (13 studies) and suppression during DSI was 51% (11 studies). Two studies quantify DSE at PF-MLI synapses and show a magnitude of 30% and 70%. Our results of about 50% suppression during DSE and DSI are thus consistent with values reported in these studies. We think that increasing n is unlikely to provide

any insight to the regulation of Ca channels at excitatory vs inhibitory synapses, nor to further our understanding of the learning-induced change in 2-AG degradation.

Line 138: We now include one sentence “The magnitude of peak suppression of DSE and DSI is consistent with previous reports”.

▲ DSE at PF-PC; ▲ at PF-MLI; ▲ our data at PF-MLI
● DSI at MLI-PC; ● our data at MLI-MLI synapse

GABA release: here the reviewer pointed out the importance of wording. If there is no evidence for change in release, please do not use it.

We have made following changes.

Page 5 line 139, from “Enhanced GABA release drives the accelerated degradation of endocannabinoids” to “GABA drives the accelerated degradation of endocannabinoids”

Line 158: from “To confirm that GABA release mediates the change in recovery rate via activation of GABA_A-receptors” to “To confirm that GABA mediates the change in recovery rate via activation of GABA_A-receptors”

Line 191: from “Given that photostimulation of MLIs and fear conditioning both increased GABA secretion and accelerated DSE recovery rate, our results suggest that a high level of GABA release is a physiological regulator that promotes 2-AG degradation.” To “Given that photostimulation of MLIs and fear conditioning both increased GABA secretion and accelerated DSE recovery rate, our results suggest that a high level of MLI activity or GABA release is a physiological regulator that promotes 2-AG degradation.”

The reviewer raised a point that the authors apparently randomly used evoked and spontaneous EPSC/IPSCs in the study. In their rebuttal, they provided explanation of why they believe that using once this and then the other is correct. The reviewer feels that they should have performed experiments consistently using the same type of postsynaptic response.

The reviewer suggests that we should have performed experiments consistently using the same type of postsynaptic response. We consistently used excitatory post-synaptic response when this was possible (Figures 1, 4 and 5). However, one characteristic feature of the mechanism we investigated (learning-induced alteration in MAGL expression), is that it affects both excitatory and inhibitory synapses regardless of whether eCB is produced by MLIs or PCs in a “synapse-independent” manner. We therefore tested this prediction using both DSE and DSI evoked by depolarizing MLIs as well as PCs in order to highlight that this unusual form of circuitry plasticity affects both excitatory and inhibitory transmission as well as neuronal excitability in cerebellar neurons.

We have made the following change in Discussion section on page 11 line 258: from “Second, in contrast to input-specific synaptic plasticity, a change in endocannabinoid degradation alters synaptic transmission in a synapse-independent manner^{4,34}, as well as neuronal excitability of inhibitory interneurons, consequently modifying the activity of the entire cerebellar circuit” to “Second, in contrast to input-specific synaptic plasticity, a change in endocannabinoid degradation alters both excitatory and inhibitory synaptic transmission in a synapse-independent manner^{4,34}, as well as neuronal excitability of inhibitory interneurons, consequently modifying the activity of the entire cerebellar circuit.”

We hope that the changes help to clarify the points that were raised concerning the learning-induced changes in DSE and DSI recovery time. We thank reviewers for these helpful suggestions.

References:

Yang, Y., Kreko-Pierce, T., Howell, R. & Pugh, J. R. Long-term depression of presynaptic cannabinoid receptor function at parallel fibre synapses. *The Journal of Physiology* 597, 3167–3181 (2019).

Wang, D.-J. et al. Cytosolic phospholipase A2 alpha/arachidonic acid signaling mediates depolarization-induced suppression of excitation in the cerebellum. *PLoS ONE* 7, e41499 (2012).

Liu, X. et al. Coordinated regulation of endocannabinoid-mediated retrograde synaptic suppression in the cerebellum by neuronal and astrocytic monoacylglycerol lipase. *Sci Rep* 6, 35829 (2016).

Zhong, P. et al. Genetic deletion of monoacylglycerol lipase alters endocannabinoid-mediated retrograde synaptic depression in the cerebellum. *J. Physiol. (Lond.)* 589, 4847–4855 (2011).

Ohno-Shosaku, T. et al. Presynaptic cannabinoid sensitivity is a major determinant of depolarization-induced retrograde suppression at hippocampal synapses. *J. Neurosci.* 22, 3864–3872 (2002).

Pan, B. et al. Blockade of 2-arachidonoylglycerol hydrolysis by selective monoacylglycerol lipase inhibitor 4-nitrophenyl 4-(dibenzo[d][1,3]dioxol-5-yl(hydroxy)methyl)piperidine-1-carboxylate (JZL184) Enhances retrograde endocannabinoid signaling. *J. Pharmacol. Exp. Ther.* 331, 591–597 (2009).

Tanimura, A., Kawata, S., Hashimoto, K. & Kano, M. Not glutamate but endocannabinoids mediate retrograde suppression of cerebellar parallel fiber to Purkinje cell synaptic transmission in young adult rodents. *Neuropharmacology* 57, 157–163 (2009).

Szabo, B. et al. Depolarization-induced retrograde synaptic inhibition in the mouse cerebellar cortex is mediated by 2-arachidonoylglycerol. *J. Physiol. (Lond.)* 577, 263–280 (2006).

Kreitzer, A. C. & Regehr, W. G. Cerebellar depolarization-induced suppression of inhibition is mediated by endogenous cannabinoids. *J. Neurosci.* 21, RC174 (2001).

Diana, M. A., Levenes, C., Mackie, K. & Marty, A. Short-term retrograde inhibition of GABAergic synaptic currents in rat Purkinje cells is mediated by endogenous cannabinoids. *J. Neurosci.* 22, 200–208 (2002).

Hashimoto-dani, Y. et al. Acute inhibition of diacylglycerol lipase blocks endocannabinoid-mediated retrograde signalling: evidence for on-demand biosynthesis of 2-arachidonoylglycerol. *The Journal of physiology* 591, 4765–4776 (2013).

Tanimura, A. et al. Synapse type-independent degradation of the endocannabinoid 2-arachidonoylglycerol after retrograde synaptic suppression. *Proc. Natl. Acad. Sci. U.S.A.* 109, 12195–12200 (2012).

Tanimura, A. et al. The endocannabinoid 2-arachidonoylglycerol produced by diacylglycerol lipase α mediates retrograde suppression of synaptic transmission. *Neuron* 65, 320–327 (2010).

Beierlein, M. & Regehr, W. G. Local interneurons regulate synaptic strength by retrograde release of endocannabinoids. *J. Neurosci.* 26, 9935–9943 (2006).

Brenowitz, S. D., Best, A. R. & Regehr, W. G. Sustained elevation of dendritic calcium evokes widespread endocannabinoid release and suppression of synapses onto cerebellar Purkinje cells. *J. Neurosci.* 26, 6841–6850 (2006).

Lebois, L. A. M., Seligowski, A. V., Wolff, J. D., Hill, S. B. & Ressler, K. J. Augmentation of Extinction and Inhibitory Learning in Anxiety and Trauma-Related Disorders. *Annu Rev Clin Psychol* 15, 257–284 (2019).

Colmers, P. L. W. & Bains, J. S. Presynaptic mGluRs Control the Duration of Endocannabinoid-Mediated DSI. *J. Neurosci.* 38, 10444–10453 (2018).

Kano, M., Ohno-Shosaku, T., Hashimoto-dani, Y., Uchigashima, M. & Watanabe, M. Endocannabinoid-mediated control of synaptic transmission. *Physiol. Rev.* 89, 309–380 (2009).

Ruediger, S. *et al.* Learning-related feedforward inhibitory connectivity growth required for memory precision. *Nature* 473, 514–8 (2011).